# Thermal infrared observations of a western United States biomass burning aerosol plume

Blake T. Sorenson[1], Jeffrey S. Reid[2], Jianglong Zhang[1], Robert E. Holz[3], William L. Smith Sr.[3], Amanda Gumber[3]

[1]Department of Atmospheric Sciences, University of North Dakota, Grand Forks, ND, 58202, United States of America
[2]Marine Meteorology Division, Naval Research Laboratory, Monterey, CA, 93940, United States of America
[3]Space Science and Engineering Center, University of Wisconsin, Madison, WI, 53706, United States of America

*Correspondence to: Blake T. Sorenson (blake.sorenson@und.edu)*

**Abstract.** Biomass burning smoke particles, due to their sub-micron particle size in relation to the average thermal Infrared (TIR) wavelength, theoretically have negligible signals at the TIR channels. However, near-instantaneous longwave (LW) signatures of thick smoke plumes can be frequently observed at the TIR channels from remotely sensed data, including at 10.6 µm (IR window) as well as in water vapor-sensitive wavelengths at 7.3, 6.8, and 6.3 µm (e.g., lower, middle and upper troposphere). We systematically evaluated multiple hypotheses as to causal factors of these IR signatures of biomass burning smoke using a combination of Aqua MODerate resolution Imaging Spectroradiometer (MODIS) and Cloud and the Earth Radiant Energy System (CERES), Geostationary Operational Environmental Satellite 16/17 (GOES-16/17) Advanced Baseline Imager, and Suomi-NPP Visible Infrared Imaging Radiometer Suite (VIIRS) and Cross-track Infrared Sounder (CrIS) data. The largely clear transmission of light through wildfire smoke in the near infrared indicates that coarse or giant ash particles are unlikely to be the dominant cause. Rather, clear signals in water vapor and TIR channels suggest both co-transported water vapor injected to the mid to upper troposphere and surface cooling by the reduction of surface radiation by the plume are more significant, with the surface cooling effect of smoke aloft being the most dominant. Giving consideration of the smoke impacts onto TIR/longwave, CERES indicates large wildfire aerosol plumes are more radiatively neutral. Further, this smoke-induced TIR signal may be used to map very optically thick smoke plumes, where traditional aerosol retrieval methods have difficulties.

## 1. Introduction

Biomass burning (BB) smoke, from both anthropogenic and natural sources such as forest fires, is and remains one of the world's dominant aerosol classes (e.g. Crutzen and Andreae 1990; Hammer et al. 2018). Significant biomass burning seasons and events occur seasonally around the globe on every continent save Antarctica. While regions of persistent burning such as central Africa and South America dominate overall emissions, perhaps most dramatic are significant mid-latitude to boreal biomass burning events becoming common in the Australia, Canada, Russia, and the United States. Recent studies have suggested that severe mid-latitude to boreal smoke events are increasing in prevalence and plumes covering larger areas (Bondur et al., 2020; Coogan et al., 2020; Phillips et al., 2022; Xian et al., 2022). BB particles are typically dominated by particles within the fine mode (volume median diameter of 0.3 – 0.6 µm) with a limited but broad coarse and giant mode of ash, dust, and other biological material that at times can be

large enough to be detected by weather radar (Reid et al., 2005; McCarthy et al., 2019). Because the fine mode is approximately 2 orders of magnitude smaller than the 10-12 µm thermal infrared (IR) wavelength regime, the aerosol optical depth (AOD, or τ) of smoke aerosol particles is often considered as negligible at thermal IR (TIR) window channels (Sutherland and Khanna, 1991). Previous studies of the radiative effects of smoke have focused mainly on

the Top of Atmosphere (TOA) radiative impacts in the shortwave (SW) spectrum (< 2.2 µm) while largely ignoring the TOA smoke aerosol radiative effects in the longwave (LW) spectrum (e.g., Chylek and Wong 1995; Christopher and Zhang 2002).

Despite the theoretical negligence of smoke particles on the TIR, easily observable BB aerosol signatures of very optically thick smoke plumes can be seen in the LW channels of weather satellite imagery. Such an example of a

dense smoke plume was provided by the NASA Aqua Moderate-Resolution Imaging Spectroradiometer (MODIS) for the 2021 Dixie Fire in northeastern California on 22 July 2021 (21:10 UTC; Fig. 1(a)-(f)). Figure 1(a) shows the Aqua MODIS true color image of the smoke plume. On this day, the Dixie Fire was in the middle of the Aqua swath, allowing for the highest resolution possible from the MODIS instrument. Remarkably, the visibly dense smoke pattern seen in Fig. 1(a) is barely noticeable in the (Fig. 1(b)) 1.24 and (Fig. 1((c)) 2.13 um channels other than through

isolated pyrocumulus with heights < 5km (Fig. 1(b) and (c)), yet the pattern closely matches an infrared cooling pattern found in the brightness temperature data from the MODIS 11 µm channel in Fig. 1(f). Within the plume region, 11 µm brightness temperatures are as much as 25 K lower than in nearby regions outside of the plume. While less significant than the 11 µm brightness temperature cooling, the MODIS (Fig. 1(d)) 3.75 µm and (Fig. 1(e)) 7.32 µm brightness temperatures also show evidence of plume-related cooling signatures.

There are several possible reasons for the observed smoke IR signals. Firstly, residual ash or entrained soil particles, which can have particle size up to 1+ mm in diameter (Reid et al., 2005; Kavouras et al., 2012), may exist in smoke plumes and introduce detectable signals at the IR spectrum. Secondly, smoke plumes have been found to contain higher water vapor mixing ratios than the ambient air due to evaporation of liquid water in the biomass and by entrainment of lower atmosphere water vapor (e.g., Clements et al. 2006, 2007; Parmar et al. 2008; Pistone et al.

2021). The elevated water vapor amount could also introduce thermal signals at the IR channels. Lastly, by reducing surface downwelling solar radiation, smoke plumes could cause surface and near-surface atmospheric cooling (Westphal and Toon, 1991; Robock, 1988, 1991; Zhang et al., 2016; Carson-Marquis et al., 2021), which may also introduce IR signals as detected from space. By using the thermal contrast between the cooled plume regions and the surrounding clear regions in MODIS 11.0 µm brightness temperature, Lyapustin et al (2020) developed a method for

deriving plume heights over dense smoke plumes from the Rocky Mountains in 2008 by attributing thermal signature absorption by entrained gas species within the plume (carbon dioxide, water vapor, nitrous oxide, methane, and ammonia) and pyrocumulus clouds; surface irradiance reduction and cooling of the surface below the plume are assumed to be negligible. Also, only the MODIS 11.0 µm brightness temperature were used in the study and no exploration was conducted for other IR channels. Clearly, there is a need to carefully study the causes of the smoke

induced TOA IR cooling at various IR channels that can be impacted by smoke aerosol.

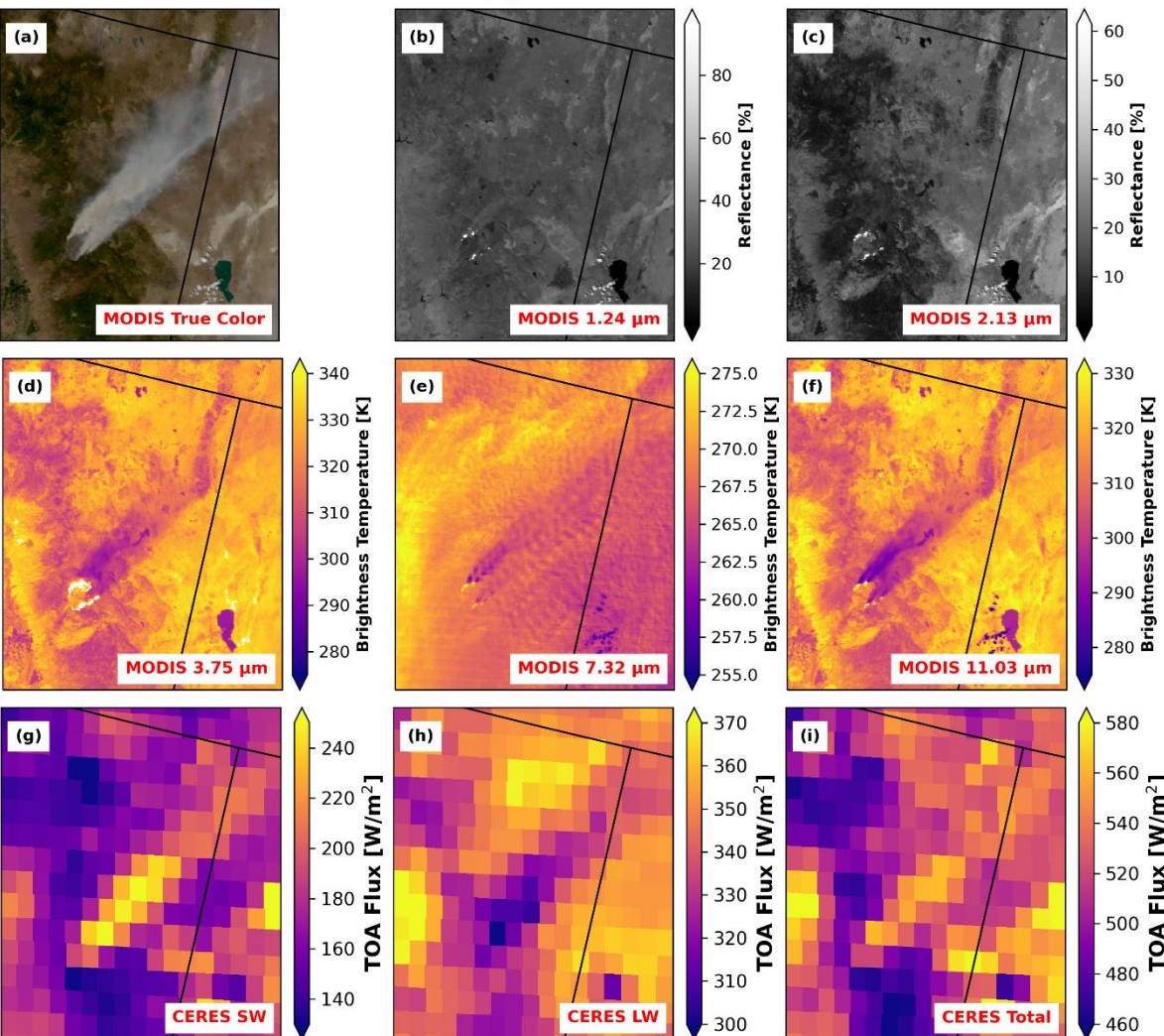

**Figure 1: Aqua MODIS and CERES data for the Dixie Fire smoke plume over NE California on 22 July 2021, 21:10 UTC. a) MODIS true color image b) MODIS channel 5 (1.24 μm) shortwave (SW) IR reflectance. c) MODIS channel 7 (2.1 μm) SWIR reflectance. d) MODIS channel 20 (3.9 μm) IR brightness temperature. e) MODIS channel 28 (7.32 μm) water vapor IR brightness temperature. f) MODIS channel 31 (11.0 μm) thermal IR brightness temperature. g) CERES TOA shortwave (SW) flux. h) CERES TOA longwave (LW) flux. i) CERES TOA total (SW + LW) flux.**

In addition to the impact of smoke on individual infrared channels, there is likely an integrated effect of smoke across the solar and terrestrial radiation spectrums. Indeed, while it is well-known that smoke plumes impact broadband radiation at the SW spectrum (Christopher and Zhang , 2002; Zhang et al., 2005), we found by examining TOA upwelling LW fluxes derived from the Aqua Cloud and the Earth Radiant Energy System (CERES) instrument, smoke plumes can also have counteracting SW and LW impacts to the TOA energy balance (e.g., Fig. 1(g) and (h), respectively). By comparing MODIS true color imagery (Fig. 1(a)) with these CERES fluxes and the overall TOA net flux (Fig. 1(i)), the net daytime flux perturbation from smoke aerosols is largely diminished. While SW fluxes within the plume region are as much as 80 Wm$^{-2}$ higher than in nearby clear-sky regions, LW fluxes in those same regions are about 50 Wm$^{-2}$ lower than in the clear-sky areas, with the resulting net daytime fluxes being only 30 Wm$^{-2}$ higher

than the surrounding areas. While the "surface dimming effect" of aerosol particles and subsequent surface cooling effect is well documented (Wild, 2009; Zhang et al., 2016; Carson-Marquis et al., 2021), this is in contrast to the use of $CO_2$, water vapor, etc. in the smoke aerosol dispersed phase to infer plume heights (e.g. Lyapustin et al., 2020), not unreasonable given the smoke's clear impact on water vapor channels. Yet, inline aerosol models with coupled

radiation typically ignore the aerosol dispersed phase contributions of entrained water vapor. This conundrum requires attribution of constituents versus surface cooling in order to understand radiative perturbations to the surface, atmosphere and TOA regimes.

From Aqua MODIS and CERES observations, it is evident that the dense smoke plumes cause observable cooling signals across multiple IR wavelengths, both in water vapor and atmospheric window channels. This may ultimately

lead to a more neutral forcing from significant biomass burning events and significant interpretation differences in surface and atmospheric heating and cooling rates. In this study, using a combination of satellite observations, near-surface air temperature measurements, and radiative transfer model simulations, we explore the origin and radiative consequences of biomass burning radiative signals in the infrared observed in the Dixie Fire BB aerosol plume from three possible mechanisms: 1) Residual ash or entrained soil particles; 2) $CO_2$ and entrained water vapor into the

smoke plume; 3) Surface cooling. Lastly, with the use of combined MODIS and CERES data, we further investigate the smoke radiative effect with smoke impacts at both SW and LW for the Dixie Fire BB aerosol plume.

### 2. Data and Methods

Satellite observations from Aqua MODIS and CERES, Geostationary Operational Environmental Satellite (GOES)-16/17, (Suomi-NPP) Visible Infrared Imaging Radiometer Suite (VIIRS) and Cross-track Infrared Sounder (CrIS)

data from 20 through 23 July 2021 are used to study the Dixie Fire BB aerosol plume. The Santa Barbara DISORT Atmospheric Radiative Transfer (SBDART) radiative transfer model is used to simulate TOA radiation as observed from MODIS and GOES for the study case. Surface observations from the Automated Surface Observing System (ASOS) data are also used from 1 July 2021 to 22 July 2021 to examine the impact of smoke to 2-m air temperature. Lastly, CrIS data are further used to study changes in vertical distributions of temperature and moisture (e.g. Smith et

al., 2015, 2021) for the study case.

### 2.1. Aqua MODIS data

The MODIS instrument is onboard both the Terra and Aqua satellites, providing spectral radiance observations at 36 channels ranging from visible to thermal IR channels (Justice et al., 1998). Aqua MODIS channels 1 (0.62-0.67 µm, 250 m resolution), 5 (1.23-1.25 µm, 500 m resolution), 7 (2.11-2.16 µm, 500 m resolution), 20 (3.66 – 3.84 µm, 1 km

resolution), 28 (7.18 – 7.48 µm, 1 km resolution) and 31 (10.78-11.28 µm, 1 km resolution) from Collection 6.1 Aqua MODIS Level 1B 1km radiance data are used.

### 2.2. Suomi-NPP VIIRS data

The Suomi National Polar-orbiting Partnership (Suomi-NPP) VIIRS provides upwelling radiance measurements across 22 channels ranging from the visible to the thermal IR channels (Lee et al., 2006). Level-1B calibrated radiances

from Suomi-NPP are used in this study to investigate the thermal characteristics of dense smoke plumes in the

overnight hours. VIIRS moderate-resolution channels 5 (0.67 µm) and 15 (10.76 µm) and the day/night band (DNB) are analyzed, all of which have 750-m spatial resolution. The VIIRS DNB uses a panchromatic wavelength range (0.5 – 0.9 µm) to measure reflected solar/lunar light on nights with at least a half-illuminated lunar disk.

### 2.3. GOES 16/17 data

Geostationary Operational Environmental Satellite (GOES) 16/17 Advanced Baseline Imager (ABI) Level 1B radiances (Schmit et al., 2017) are used to study the temporal variation of the smoke plume and its impacts on the Dixie Fire case. ABI also has the advantage of multiple water vapor channels. The GOES-16/17 ABI provides scans of the CONUS domain every 5 minutes across 16 channels, ranging from the visible to the TIR (Schmit et al., 2017), with GOES-16/17 Level-1B radiances being studied for the Dixie Fire case. GOES-16/17 radiances from the contiguous United States (CONUS) scan from channels 2 (0.64 µm, spatial resolution 0.5 km), 6 (2.2 µm, spatial resolution 2 km), 7 (3.9 µm, spatial resolution 2 km), 8 (upper-level water vapor, 6.19 µm, spatial resolution 2 km), 9 (mid-level water vapor, 6.95 µm, spatial resolution 2 km), 10 (lower-level water vapor, 7.34 µm, spatial resolution 2 km), and 13 (clean IR longwave window 10.35 µm, spatial resolution 2 km) at 3-hour intervals between 12:00 UTC 20 July 2021 and 03:00 UTC 22 July 2021 were provided by the University of Wisconsin-Madison Space Science and Engineering Center (SSEC). Additional GOES-16/17 CONUS scans for every 5 and 30 minutes throughout the study period were separately accessed from the Amazon Web Services (AWS) online data bucket at https://noaa-goes17.s3.amazonaws.com/index.html and https://noaa-goes16.s3.amazonaws.com/index.html.

### 2.4. Suomi-NPP CrIS data

The Suomi-NPP Cross-track Infrared Sounder (CrIS) is a hyperspectral spectrometer that measures upwelling radiances across 1305 channels in the longwave IR, midwave IR, and shortwave IR, allowing for the retrieval of atmospheric temperature and moisture profiles (Han et al., 2013). Clear-sky retrievals of surface skin temperature as well as profiles of atmospheric air temperature and mixing ratio (Smith et al., 2015, 2021) within and surrounding the Dixie Fire smoke plume are analyzed in this study.

### 2.5. CERES data

The Cloud and the Earth Radiant Energy System (CERES) instrument, on board the Terra, Aqua, and NOAA-20 satellites, measures broadband radiance at the SW and total spectra, which are further used for estimating TOA SW and LW fluxes using predefined angular distribution models (Su et al., 2015; Kratz et al., 2014). The spatial resolution for each CERES pixel is on the order of 10 km at nadir. The CERES single scanner footprint (SSF) data, which contains collocated MODIS (aerosol and cloud) and CERES data, are used in this study to quantify how the thick aerosol plumes and associated TIR brightness temperature reduction relate to TOA upwelling radiation. To collocate the MODIS radiance and CERES data in this study, all MODIS pixels within 0.1 degrees latitude and longitude of each CERES pixel are averaged.

### 2.6. ASOS data

The 2-m air temperature data from two ground stations in northeastern California are used to study the surface temperature effects of the dense smoke plume on 22 July 2021: O05 (Chester, California) and AAT (Alturas, CA).

ASOS data from O05 and AAT stations were downloaded from the Iowa State University ASOS database ( https://mesonet.agron.iastate.edu/request/download.phtml?network=AWOS) for the period between 1 July 2021 and 23 July 2021. While station O05 was covered with smoke on the study days (21 – 22 July 2021), station AAT, which is 146 km northeast of O05, is free from heavy smoke on 22 July 2021. The ASOS data from the beginning of the month are used to develop a baseline diurnal cycle for each station to compare with the cycles from the smoky day of 22 July 2021. The 'baseline' period is chosen to be 1 July 2021 to 13 July 2021 because the Dixie Fire had not yet started (and thus there was no smoke), and through visual inspection of satellite imagery it was determined that there were very few clouds over the region over this period, so the diurnal cycles from each day within this period were unperturbed by clouds and smoke.

While there are several other ASOS stations near O05 (CIC, RBL, OVE, and SVE), these sites are not included in the study for several reasons. First, due to the fact that the study region in northeastern California covers the northern ends of both the Sierra Nevada mountains and California's central valley, there are significant elevation differences across the study region. While ASOS stations CIC, RBL, and OVE are close to the station of interest (O05), these stations are located in the central valley at elevations nearly 1 km below O05. These elevation discrepancies cause changes to the diurnal temperature cycles at the stations and complicate attempts to study changes to the diurnal temperature cycles caused by thick smoke plumes. Second, while ASOS station SVE is very close to O05, it suffered from extended reporting outages during the study period, especially on 22 July 2021. A large portion of the outage on 22 July 2021 took place during middle and late afternoon hours, precisely the time when the overhead smoke plume at O05 was the thickest and caused the strongest cooling at O05, making any temperature comparisons between the two stations impossible. The ASOS station in Alturas, CA (AAT), northeast of station O05, is at a similar elevation as O05 and had no data quality issues during this period, and thus is also used.

### 2.7. NEXRAD data

To assist with determining the impacts of possible large BB smoke particle and/or hydrometeors on the observed TIR signal, we analyze NOAA Next Generation Radar (NEXRAD) data in the smoke plume region. NEXRAD is a network of 160 Weather Surveillance Radar -1988 (WSR-88D) 10-cm wavelength radars spread across the United States and its military installations (NOAA National Weather Service (NWS) Radar Operations Center, 1991). Level II NEXRAD data files containing observations of reflectivity and cross correlation ratio from the Beale Air Force Base (KBBX, southwest of the Dixie Fire) and Reno, NV National Weather Service (KRGX, southeast of the Dixie Fire) WSR-88D radars for select times between 20 July 2021 and 23 July 2021 are obtained from the Amazon Web Services NEXRAD data repository (https://s3.amazonaws.com/noaa-nexrad-level2/index.html). Composite reflectivity is defined as the maximum radar reflectivity observed across any of the elevation angles over a given point, and is used in this study to identify the general location of radar returns within the entire smoke plume region. Cross correlation ratio (or correlation coefficient) is a measure of the similarity of the power returned in the vertically- and horizontally-polarized pulses, and can be used to identify the uniformity of targets within a radar volume. High correlation coefficient (0.95 – 1.00) is associated with uniform meteorological targets (pure snow or pure rain), while low correlation coefficient (< 0.8) is associated with non-meteorological targets (i.e. birds, bugs, and ash). Analysis and

visualization of the NEXRAD radar data are conducted using the Python ARM Radar Toolkit (Py-ART) (Helmus and Collis, 2016).

## 2.8. SBDART model

To investigate the differences in radiance observed by the GOES-16/17 water vapor channels and the GOES-16/17 and MODIS thermal IR channels caused by changes in water vapor mixing ratio, simulated TOA radiances are calculated using the Santa Barbara DISORT Atmospheric Radiative Transfer (SBDART) model (Ricchiazzi et al., 1998). The SBDART model is used to simulate the Aqua MODIS and GOES-16/17 observed TOA radiances as a function of viewing zenith angle for different atmospheric conditions, with the atmospheric profiles for the SBDART runs being extracted from Suomi-NPP CrIS temperature and humidity retrievals. Filter functions from various GOES and MODIS satellite channels as used are included in the SBDART model for simulating filtered radiances as observed from satellites. These filter functions are obtained from the EUMETSAT Numerical Weather Prediction (NWP) Satellite Applications Facilities (SAF) online archive at https://nwp-saf.eumetsat.int/site/software/rttov/download/coefficients/spectral-response-functions/.

## 3. Dixie Fire BB plume TOA IR cooling sources

As mentioned in the introduction section, there are a several possible reasons for the smoke aerosols to be observable at the TIR channels. Here we systematically examine the leading contenders in the context of the 2021 Dixie Fire case: 1) Impact of the presence of coarse or giant aerosol particles in the smoke or ice crystal formations for particularly high injections that may impact light extinction in the infrared; 2) The role of injected water vapor found in free-tropospheric smoke plumes (e.g. Pistone et al. 2021); and 3) the reduction in downwelling solar radiation reaching the surface response of soil temperature (e.g., Zhang et al. 2016; Carson-Marquis et al. 2021).

### 3.1. Co-emitted coarse and giant particles

We first study the possible impacts of coarse BB particles and/or pyrometeors on the observed TOA IR signal, with pyrometeors defined by McCarthy et al. (2019) as pyrogenic debris greater than 1 mm in diameter. An effective way of estimating the size of aerosol particles from multispectral remotely sensed observations is with the Angstrom exponent, which defines the change in the optical depth of an aerosol species with respect to the wavelength of the incident light, is related to the volumetric mean aerosol particle size by means of:

$$\tau_a = \tau_0 \lambda^{-\alpha} \tag{1}$$

where $\tau_a$ is the optical depth at wavelength $\lambda$ (in μm), $\tau_0$ is the optical depth at a reference wavelength ($\lambda = 1$ μm), and $\alpha$ is the Angstrom exponent (Angstrom, 1929). A small Angstrom exponent value (e.g. $< 1$) indicates that the AOD for a certain aerosol species does not vary significantly with increasing wavelength, while large Angstrom exponent values (e.g. $> 1$) indicate that the AOD of an aerosol species varies significantly with increasing wavelength. For example, coarse mode particles such as dust have an Angstrom exponent value of less than 1 at the visible spectrum, but fine-mode particles such as smoke and anthropogenic fine-mode aerosols have large Angstrom exponent values (e.g. ~1.5-2 at the visible spectrum). AOD of those fine-mode aerosols decreases significantly from smaller wavelengths (visible, ~ 0.64 μm) to larger wavelengths (SWIR, ~ 2 μm) (Westphal and Toon, 1991; Eck et al., 1999).

Thus, we can use these concepts to examine if large BB particles are responsible for the strong TIR cooling signals seen in the smoke plume. If the observed visible and SWIR reflectances in the dense smoke plume are both very high, this suggests a small Angstrom exponent value and, therefore, large (> 0.5 µm) particles are widespread in the smoke plume. However, if the observed SWIR reflectance in the plume is much lower than the visible reflectance, this suggests a large Angstrom exponent value and, therefore, the plume consists primarily of small, fine-mode (< 0.5 µm) aerosol particles (Schuster et al., 2006).

We examine MODIS 2.1 µm reflectance, shown in Fig. 1(c), to determine if large ash particles or pyrocumulus clouds are present in the 22 July 2021 BB plume and are contributing to the TIR cooling signal. In an area of the plume with visibly dense smoke, the MODIS 0.64 µm reflectance is an average of 20 percentage points higher than just outside the plume, with an average 11 µm brightness temperature difference of 21 K between the same areas. However, the average difference in MODIS 2.1 µm reflectance between those areas is only a statistically insignificant 0.8 percentage points, suggesting that no relationship exists between the 2.1 µm reflectance and the TOA cooling and that smoke AOD / pyrocumulus cloud optical depth at the 2.1 µm channel is nearly negligible. With the SWIR reflectance in the plume being significantly lower than the visible reflectance in the plume, the data suggest that the aerosols in the smoke plume have a large Angstrom exponent, which indicates that the smoke plume is dominated by fine-mode smoke aerosols. Therefore, it is concluded that the smoke signature seen in the TIR channel for the 22 July 2021 case is not caused by large debris or pyrocumulus generated from the BB event.

While we observed a plume TIR cooling effect in MODIS without ash effects, this does not mean that ash effects are not universally a contributor, just not in this particular case. For example, the Dixie Fire BB aerosol plume one day prior (21 July 2021, 00:00 UTC, Fig. 2) also induces an IR cooling pattern, as illustrated using GOES-17 data (GOES-17 was offline on 22 July 2021 due to a "satellite anomaly and ABI reset", https://www.star.nesdis.noaa.gov/GOESCal/goes_SatelliteAnomalies.php). The GOES-17 10.35 µm brightness temperatures, shown in Fig. 2f, reveal cooling of up to 25 K in the smoke plume region, similar in magnitude to the cooling found in the MODIS TIR brightness temperature data from 22 July 2021. In addition to the observed TIR cooling, plume signals are also observable in GOES-17 water vapor channels (Fig. 2(d)-(f)). For the smoke plume shown in the GOES-17 visible image (Fig. 2(a)), the GOES-17 upper-level and mid-level water vapor brightness temperatures (Fig. 2(d) and (e)) show localized areas of TOA cooling downwind of the fire, with brightness temperatures in these localized regions about 5 K and 8 K cooler than regions just outside the plume region for the upper- and mid-level water vapor channels, respectively. However, the cooling pattern seen in the 10.35 µm data is not visible in either the upper- or mid-level water vapor channels. The GOES-17 low-level water vapor brightness temperatures (Fig. 2(f)) show a combination of both cooling patterns: both the localized cooling areas seen in the upper- and mid-level water vapor imagery and the plume-parallel cooling pattern seen in the thermal IR imagery are present. Within the locally enhanced cooling seen northeast of the fire, the brightness temperature cooling is approximately 10 K, while the plume-parallel cooling pattern closer to the fire exhibits a cooling of roughly 4 K

relative                              to                              the                              surroundings.

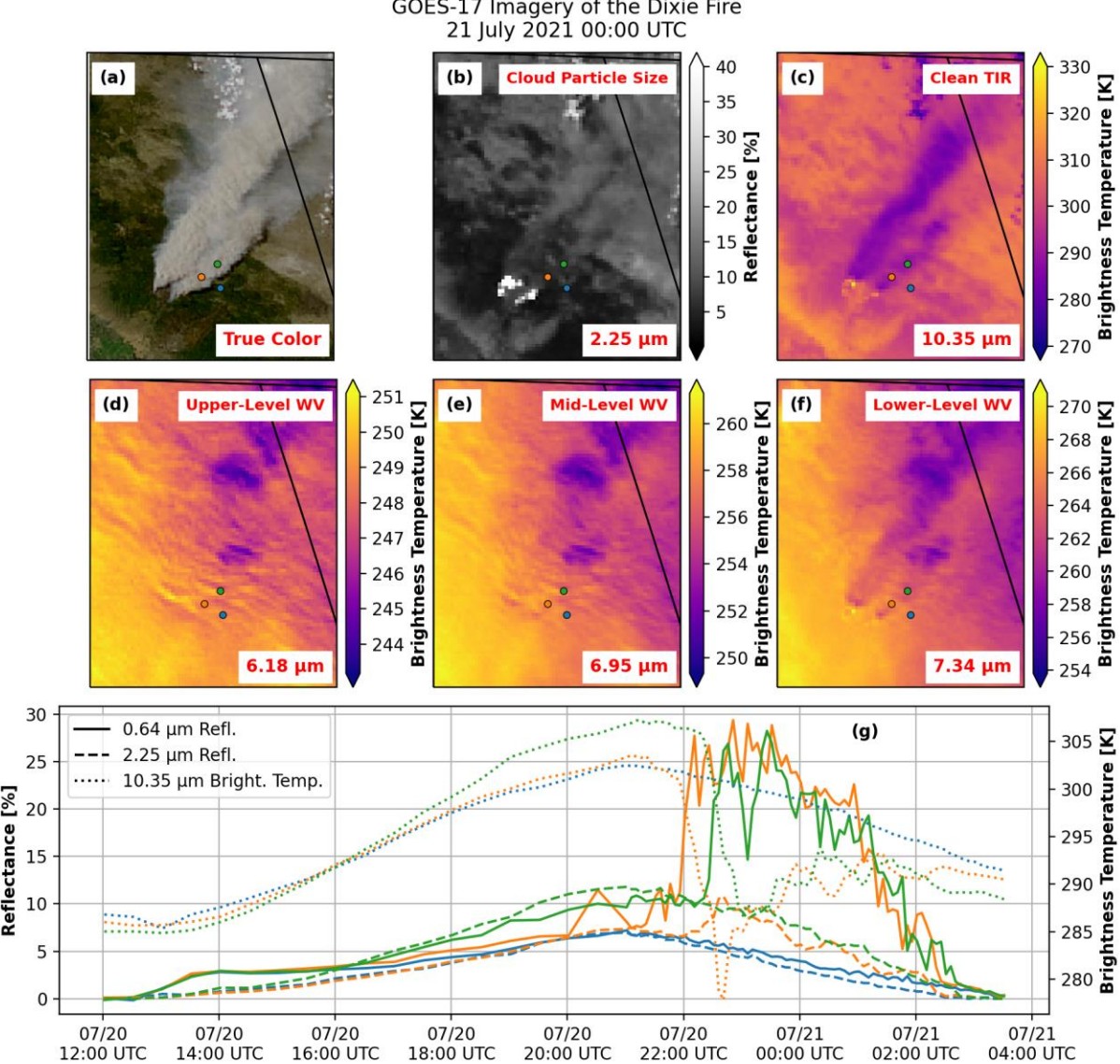

**Figure 2: GOES-17 true color (a), shortwave infrared (2.25 μm, (b)), thermal infrared (10.35 μm, (c)), upper-level water vapor (6.18 μm, (d)), mid-level water vapor (6.95 μm, (e)), and low-level water vapor (7.34 μm, (f)) imagery of the Dixie Fire at 21 July 2021, 00:00 UTC. Third row: Time series of GOES-17 0.64 μm visible reflectance (solid), 2.25 μm shortwave infrared reflectance (dashed), and 10.35 μm brightness temperature (dotted) for points outside of the Dixie Fire smoke plume (blue) and inside the plume (orange and green) near the fire site.**

Scanning the GOES-17 dataset, weak residual plume signals are sometimes observable from the SWIR channel when plumes are extremely optically thick, perhaps during fire flare ups (as noticeable in Fig. 2(g)). For example, Fig. 2(g) shows the time series of GOES-17 visible and SWIR reflectance and TIR brightness temperature for locations on the southern edge of the Dixie Fire plume on 20 and 21 July 2021. For the two selected locations with one within the Dixie Fire plume (Fig. 2(a) and (g), orange) and another outside the plume (Fig. 2(a) and (g), blue), the visible (Fig. 2(g), solid lines) and shortwave IR (Fig. 2(g), dashed lines) reflectance, as well as the TIR brightness temperatures (Fig. 2(g), dotted lines), are nearly identical at the two locations during the first half of July 20[th] when both points

were under clear skies. During the second half of the day, when the orange site was covered by the plume, the visible reflectance at the orange site increased to a maximum of 28% while the reflectance at the blue site decreased from a maximum of 7%, and the TIR brightness temperature at the orange site was as much as 15 K cooler than at the blue site, with a brief, stronger dip in TIR brightness temperature at the orange point at about 22:30 UTC caused by a short-lived pyrocumulus cloud (not shown). While the visible reflectance and TIR brightness temperature change drastically
within the plume, the SWIR reflectance exhibits changes of less than 5%. This residual plume signal, however, cannot be observed at some other locations within the plume as shown in green colors in Fig. 2(a) and (g). While about the same increase in visible reflectance and reduction in TIR brightness temperature are found at both the orange and the green locations, the residual SWIR plume signal is significantly smaller at the green point than at the orange point, showing that the TIR cooling is not primarily driven by the residual SWIR signal. Nevertheless, some residual plume
signal in the GOES-17 SWIR reflectance is not surprising because smoke AOD at ~2 µm is still around 1-6% of AOD at 0.55 µm (e.g. Levy et al. 2007; Remer et al. 2005), which may be non-negligible for very optically thick plumes with visible AOD of above 5. This, however, cannot totally exclude the possibility that some regions may be polluted with large smoke debris.

      As a final test of the potential impacts of pyrometeors and hydrometeors on the observed TIR cooling signal, we
compare WSR-88D radar observations of the plume region to the GOES-17 observations. Two nearby NOAA WSR-88D radars, KBBX (Beale Air Force Base, southwest of the Dixie Fire) and KRGX (Reno, Nevada, southeast of the Dixie Fire) provided good coverage of the smoke plume area, so horizontal and vertical cross sections of reflectivity and correlation coefficient at 2021 July 21 00:00:00 UTC are analyzed and shown in Fig. 3. Three cross sections of the radar data are taken through the smoke plume, with the cross sections plotted over the GOES-17 visible reflectance,
SWIR reflectance, and TIR brightness temperature in Fig. 3(a), (b), and (c). The plan position indicator (PPI) of reflectivity from KBBX (Fig. 3(d)) shows regions of high (> 20 dBZ) reflectivity in the regions of the smoke plume immediately downwind of the fire, but extending no more than 20 km downwind of the beginning of the plume. The KRGX PPI reflectivity (Fig. 3(g)) shows high reflectivity in similar regions near the fire, but also has regions of low reflectivity extending farther downwind than the KBBX observations show. Range height indicator (RHI) cross
sections of reflectivity through the plume region along the 36-degree azimuth from KBBX (Fig. 3(e) – (f)) show a column of high reflectivity (maximum of 40 dBZ) and very low correlation coefficient (< 0.6) centered about 70 km away from the radar and extending up to 6 km above sea level, with moderate reflectivity and slightly higher (~0.7) correlation coefficient observed at lower heights to about 90 km away from the radar. The high reflectivity and low correlation coefficient suggest the presence of pyrometeors in this region of the plume.

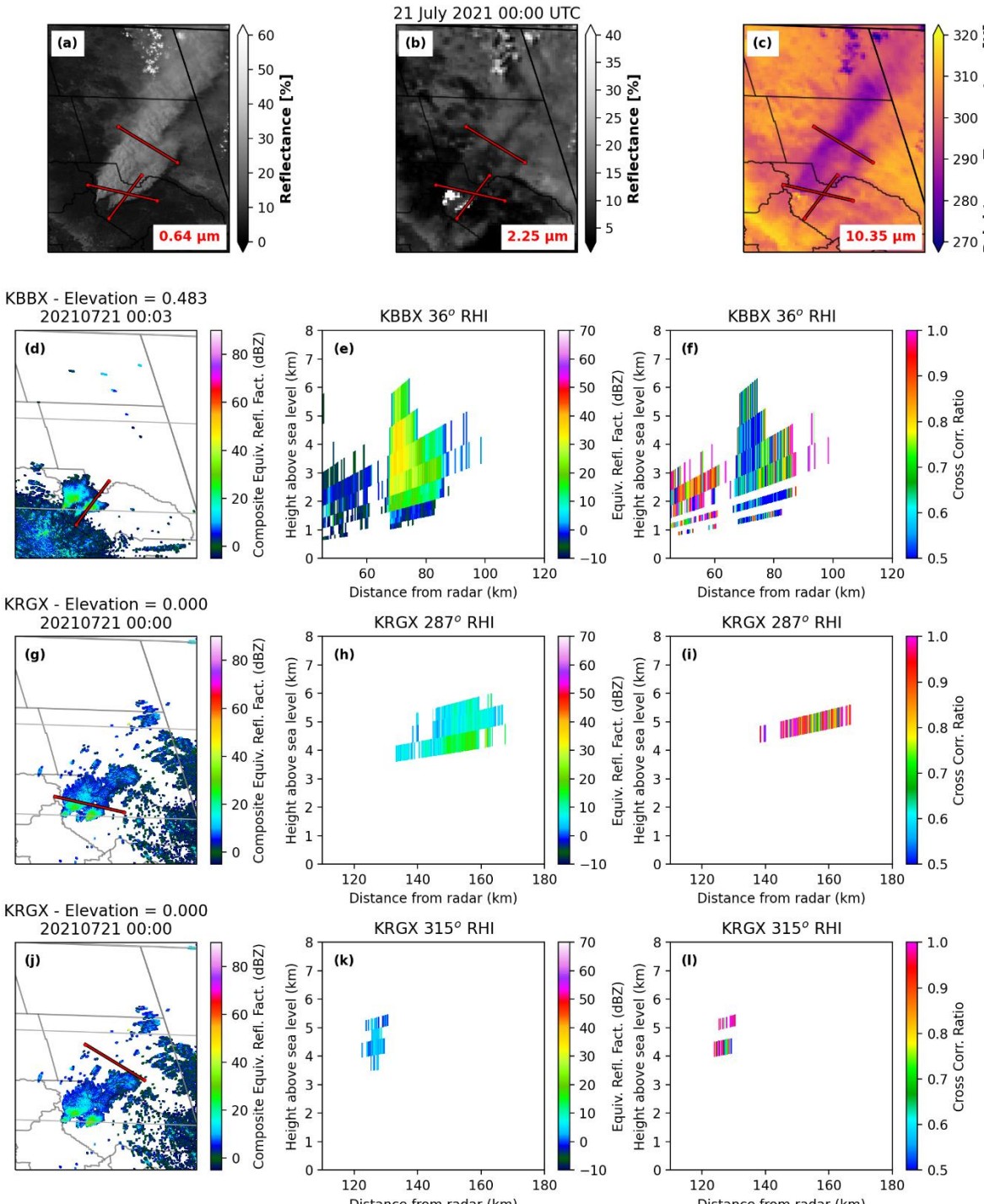


**Figure 3. Comparison of GOES-17 and NEXRAD radar observations derived from the Beale AFB radar (KBBX, southwest of figure) and Reno, NV NWS radar (KRGX, southeast of figure) at 00:00 UTC 21 July 2021. First row: GOES-17 visible reflectance (a), shortwave IR reflectance (b), and thermal IR brightness temperature (c), with radar cross section locations added as red lines along azimuths from KBBX and KRGX. Second row: KBBX plan position indicator (PPI) of composite**

**reflectivity (d), and range-height indicator (RHI) of reflectivity (e) and correlation coefficient (f). Third row: as in the second row, but for KRGX. Fourth row: as in the third row, but for a cross section much farther downwind of the fire.**

However, a cross section of the plume from KRGX very far downwind of the plume (Fig. 3(j) – (l)), in regions that the GOES-17 visible reflectance shows large amounts of smoke and the GOES-17 TIR brightness temperatures show strong cooling, show next to no reflectivity. The same magnitudes of TIR cooling are observed far downwind of the fire, where there are no radar returns, and very close to the fire, where there are significant radar returns. While the KRGX radar is at a much higher elevation than the KBBX radar (2950 m AGL for KRGX, 67 m AGL for KBBX) and thus may not see large ash and/or pyrometeors below the radar level, even the KBBX radar does not observe any returns far downwind of the fire, as indicated by both the KBBX PPI and RHI diagrams.

While the KBBX and KRGX reflectivity and correlation coefficient observations suggest the possible presence of pyrometeors in the plume in close proximity to the fire (although we also cannot rule out the impacts of Bragg scattering on the radar signal; Richardson et al., 2017) the GOES-17 SWIR reflectances do not show significant increases in reflectance in those same regions, which would be expected if large (> 0.5μm) particles were present in the plume. We thus cannot conclusively state if pyrometeors were present in large amounts across the plume region. Regardless, with the same magnitude of strong TIR cooling being observed in regions with no radar reflectivity and with high reflectivity, we conclude that pyrometeors and hydrometeors are not the primary cause of the TIR cooling signal.

### 3.2. Co emitted/transported water vapor and other gas species

Smoke plumes generated from biomass burning events have been found to contain higher water vapor mixing ratios than the ambient air (e.g., Clements et al. 2006, 2007; Parmar et al. 2008; Pistone et al. 2021). With burning of biomass, liquid water inside biomass is evaporated, water vapor is produced as a product of combustion, and higher near surface water vapor air are all entrained into the plume injected into the troposphere. The enhanced smoke plume water vapor amount can be observed in the Dixie Fire plume from water vapor mixing ratio fields as retrieved using Suomi-NPP CrIS data (e.g. Smith et al., 2015). Only retrievals from clear-sky CrIS spectral radiances are shown. The retrieved water vapor mixing ratio values at 500 mb in the Dixie Fire smoke plume in the 22 July 2021, 21:21 UTC CrIS granule (Fig. 4(c)) are about 0.6 – 0.7 g/kg, much higher than the water vapor mixing ratios of less than 0.2 g/kg found in the smoke-free environment around the plume. The region of enhanced water vapor mixing ratio closely matches the plume region indicated in the Suomi-NPP VIIRS imagery found in Fig. 4(a). This locally enhanced mixing ratio in the plume can also be seen in CrIS mixing ratio profiles (Fig. 4(e)) taken in three points: one within the plume (blue in Fig. 4), and two outside the plume (orange and green in Fig. 4). The water vapor mixing ratios at the smoky point are higher than at the clear points in the upper portions of the profile, from about 600 mb to 350 mb. However, below 600mb, water vapor mixing ratios at the smoky point are lower than at the clear points in the upper portions of the profile, which may be related to reduced air temperature as shown in Fig. 4(f).

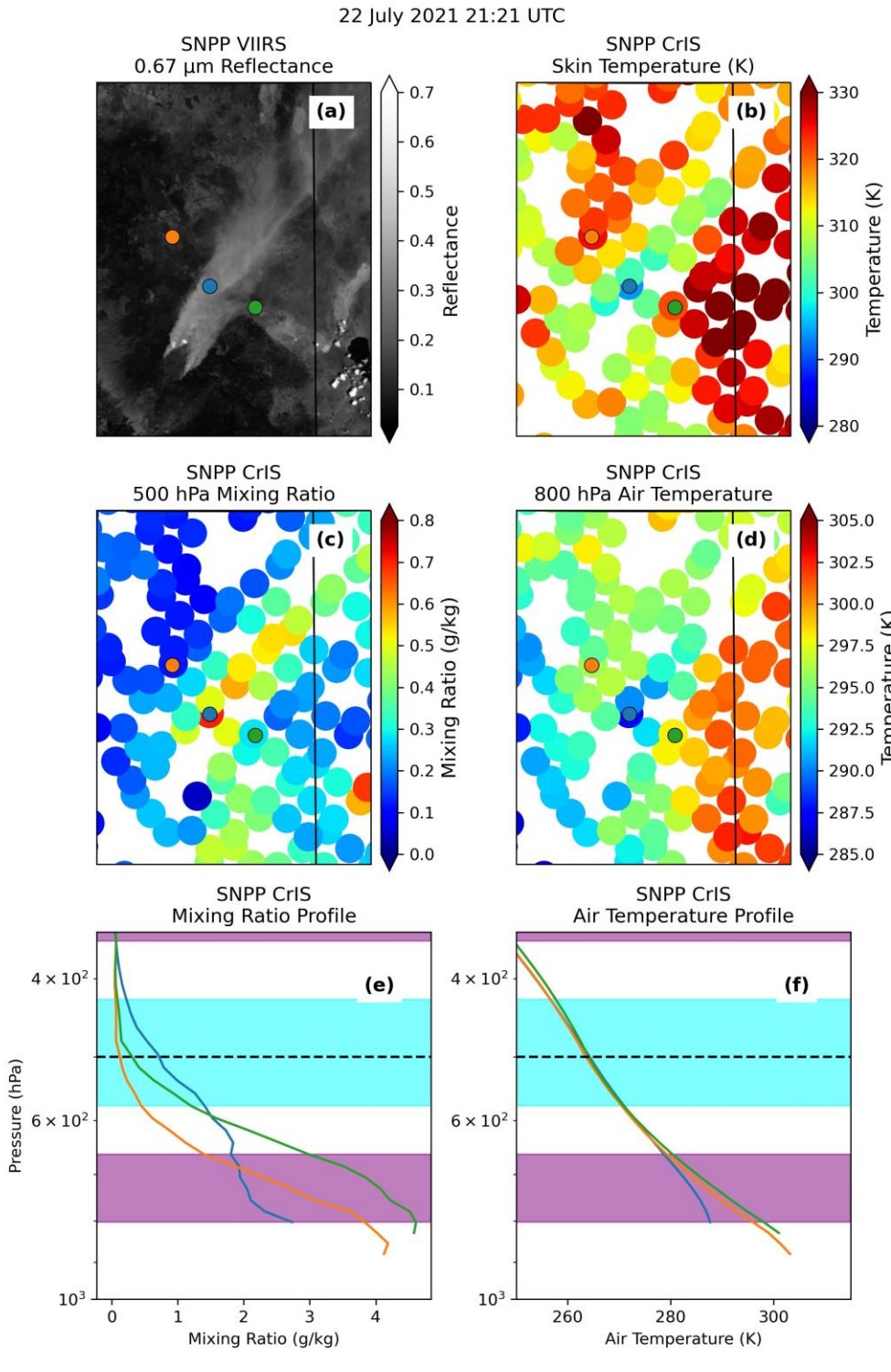

**Figure 4: Suomi-NPP VIIRS 0.67 micron visible reflectance (a) , CrIS surface skin temperature (b), CrIS 500 hPa mixing ratio (c), and CrIS 800 mb air temperature (d), as well as profiles of CrIS mixing ratio (e) and temperature (f) from one location inside the smoke plume (blue) and two locations outside of the plume (orange and green). The cyan shading in panels e and f denotes where the mixing ratio at the blue point is greater than at the orange and green points by more than 0.1 g/kg, while the purple shading denotes regions in which the air temperature at the blue point is less than at the orange and green points by more than 0.5 K.**

To explore the impact of co-transported water vapor and gas species on IR signals, both radiative transfer modeling and satellite observations are applied. For radiative transfer model runs, GOES-16 and Aqua MODIS IR brightness temperatures are simulated as functions of viewing zenith angle using SBDART, with the CrIS surface and profile data from the clear (solid orange in Fig. 4(e) and (f)) and smoky (solid blue in Fig. 4(e) and (f)) points analyzed above serving as the input data for the simulations. The GOES-16 data are used here because GOES-17 experienced a

"satellite anomaly and ABI reset" on July 22, 2021 (see the GOES ABI Calibration Events Log at https://www.star.nesdis.noaa.gov/GOESCal/goes_SatelliteAnomalies.php), and thus no GOES-17 data are available during the CrIS overpass on that day. The GOES-16 imagery at the time of the CrIS overpass is shown in Fig. 5(a)-(e), and the smoke IR signals are clearly visible in the true color imagery in Fig. 5(b), even with a ~65° viewing angle. The SBDART simulations are performed at three GOES-16 water vapor channels, as well as the GOES 10.35 µm and

MODIS 11.0 µm channels. For each selected channel, the baseline simulation, which is assumed to be smoke free, is performed using the CrIS retrieved temperature and moisture profiles and surface temperature from the clear point. The impacts of smoke aerosols due to water vapor are then simulated by replacing the moisture profile from clear point with the CrIS retrieved profile from the smoky point, with all other parameters remaining unchanged. This exercise is illustrated in Fig. 5(f)-(j), which shows the simulated GOES-16 and MODIS brightness temperatures, with

Fig. 5(f), (g), and (h) showing the simulated GOES-16 upper (channel 8, 6.2 µm, with peak response at about 340 mb), mid-level (channel 9, 6.94 µm, with peak response at about 440 mb), and lower-level (channel 10, 7.34 µm, with peak response at about 620 mb) water vapor channels, respectively, and Fig. 5(i) and (j) showing the simulated GOES 10.35 µm and MODIS 11.0 µm brightness temperatures. The vertical dashed lines in Fig. 5(f)-(i) show the viewing zenith angle of the GOES-16 data for the study case, and the vertical dashed line in Fig. 5(j) shows the viewing zenith

angle of Aqua MODIS. When substituting the mixing ratio profile from the smoky (blue) point into the profile for the clear (orange) point, the resulting simulated GOES-16 upper-level, mid-level, and low-level water vapor channel brightness temperatures at the GOES-16 viewing zenith angle (green in Fig. 5(f)-(h)) are cooler than those in the baseline run (orange in Fig. 5(f)-(h)) by 1.5, 1.5 and 5 K, respectively. Those numbers are in the ballpark of, but higher than the differences between the brightness temperature observed at the clear and smoky points as seen in the GOES-

16 data (Fig. 5(b)-(e)), with observed differences of 0.7, 0.2, and 1.5 K for the upper-, mid-, and lower level water vapor channels, respectively.

    Although changes in water vapor amount affect GOES brightness temperatures from the selected water vapor channels, the impact of water vapor on the IR window channels (GOES 10.35 µm and MODIS 11.0 µm) are rather marginal. When the water vapor profile from the smoky point is used, only slightly changes in both the GOES 10.35

µm (Fig. 5(i)) and MODIS 11.0 µm (Fig. 5(j)) brightness temperatures of less than 0.1 K are found. To further examine the impact of water vapor on brightness temperature as observed from the IR window channels, we double the water vapor mixing ratio profile at the smoky point from the surface to a height of 8 km, which is approximately the highest level in which the elevated plume mixing ratios are observed in the CrIS data. The simulated GOES-16 water vapor brightness temperatures decrease with the doubled water vapor mixing ratio, with the lower-level water vapor channel

showing the strongest cooling (about 10 K near nadir) and the upper-level water vapor channel showing the smallest cooling (about 5 K near nadir). These differences in brightness temperatures are much larger than those observed from

the GOES-16 data (Fig. 5(b)-(e)), indicating that upon doubling the water vapor mixing ratio, we greatly exceeded the actual water vapor amount in the atmosphere over the smoky point. Even with double the water vapor amount, the resulting simulated TIR brightness temperatures are only less than 5 K below the baseline values for the IR window channels, which is much smaller than the observed ~20-25 K difference from either MODIS (Fig. 1(f)) or GOES-16 (Fig. 5(i)).

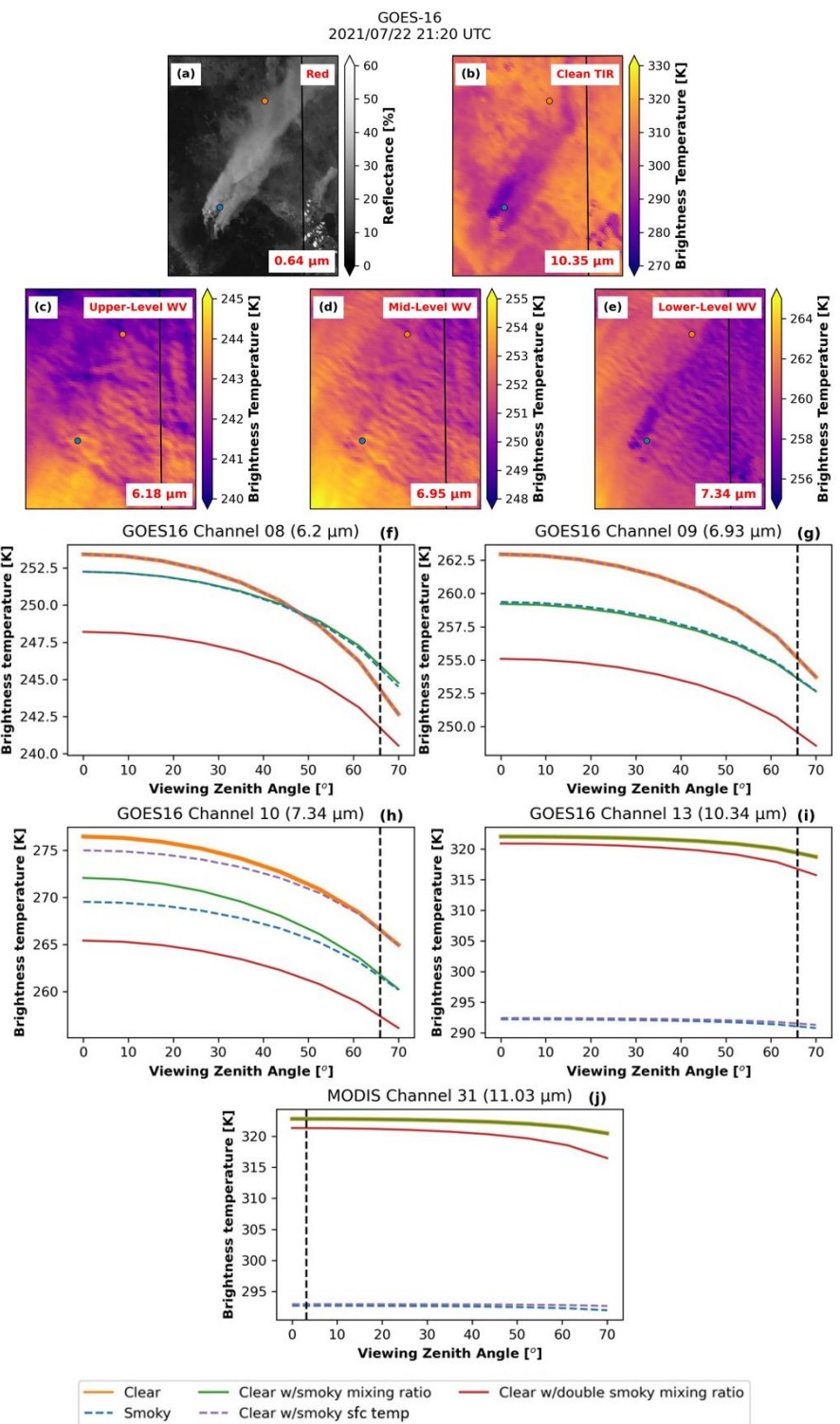

**Figure 5: GOES-16 true-color imagery (a) and brightness temperatures from the thermal IR (10.35 μm, b), upper-level water vapor (6.2 μm, c), mid-level water vapor (6.94 μm, d), and lower-level water vapor (7.34 μm, e) channels, as well as SBDART-simulated TOA brightness temperatures for the GOES-16 upper-level water vapor (f), GOES-16 mid-level water vapor (g), GOES-16 lower-level water vapor (h), thermal IR (i), and Aqua MODIS thermal infrared (11 μm, j) channels temperatures using the Suomi-NPP CrIS-retrieved atmosphere. The orange and blue dots in panels b – e display the locations of the "clear" and "smoky" points for the simulations, while the dashed lines in panels f – j represent the GOES-16 (22 July 2021, 21:20 UTC) and MODIS (22 July 2021, 21:10 UTC) viewing zenith angles for the Dixie Fire.**


Lyapustin et al (2020) suggested the source of the plume IR cooling signal to be absorption by entrained gases including carbon dioxide, nitrous oxide, ammonia, and methane, in addition to the enhanced water vapor from combustion, so we also simulate Aqua MODIS and GOES-16 TOA brightness temperatures for enhanced mixing ratios of each of these gas constituents. Given that methane and nitrous oxide do not absorb in either the 10.35 μm or 11.0 μm bands, we do not expect either of these gas species to impact the observed TIR signals in those bands, while

we could expect to see some limited impacts from ozone, carbon dioxide, and ammonia because they have some absorption lines near the TIR spectrum (Liou, 2002). For the carbon dioxide simulations, with the SEAC4RS campaign (Toon et al., 2016) observing carbon dioxide mixing ratios of up to 480 ppm within biomass burning smoke plumes, we simulated the MODIS and GOES-16 brightness temperatures with carbon dioxide mixing ratios of 500 ppm and 1000 ppm. For the gas species simulations with nitrous oxide, ammonia, and methane, since we have no observations

of the changes to the gas species mixing ratio in the smoke plumes, we double and triple the default mixing ratios of the total column gas species mixing ratio in the model to determine if they affect the simulated brightness temperatures. While not shown, the radiative transfer model simulations with enhanced mixing ratios of carbon dioxide, nitrous oxide, ammonia, and methane did not yield any significant reduction in the simulated MODIS 11.0 μm brightness temperatures; increasing the total column carbon dioxide mixing ratio to 1000 ppm reduced the 11.0 μm brightness

temperatures by only 0.2 K, while the simulations with increased nitrous oxide, ammonia, and methane resulted in negligible changes to the 11.0 μm brightness temperatures. Similar results are also found for the GOES-16 10.35 μm channel. With the exception of the GOES-16 low-level water vapor channel, which exhibited a cooling of about 1.2 K only for a total column methane mixing ratio increased from 1.74 to 3.74 ppm, none of the GOES-16 water vapor channels exhibited any response to the enhanced column gas species concentrations. Note that high concentrations of

hydrogen cyanide (HCN) were found for the 2015 Indonesian Fires (Park et al., 2021). However, no observational HCN concentration data are available to confirm the presence of high concentrations of HCN for this study case. Also, if absorption by HCN within the smoke plume plays a significant role in the TIR cooling signal for this study case, we would observe cooling signals within the smoke plume at night, but as we show later in Section 3.4, no significant cooling signal is observed in the plume region at night. We thus expect the impact of HCN to be marginal in this case,

but leave further analysis of the impacts of HCN on the Dixie Fire smoke plume to a future study.

### 3.3. Surface radiative response

Finally, to test the impact of the smoke plumes on surface conditions, we compare 2-m temperatures measured inside and outside of the thick plume region during the 22 July 2021 BB plume case. Fig. 6(a) shows the Chester, California ASOS site (O05), which was under very dense smoke during the daytime of 22 July 2021, and the Alturas, California

ASOS site (AAT), located in much thinner smoke to the northeast. The similar elevations between the two stations (1382.1 m for O05 versus 1334.5 m for AAT) reduces the amount of topographic temperature bias between the stations, and similar diurnal temperature variations over non-smoky days (Fig. 6(b), solid lines). To quantify the impact of background meteorological variation on the 2-m temperatures measured at each site, we calculated the average of the diurnal temperature cycles at each station between 1 July 2021 and 13 July 2021, days that were

primarily cloud-free and before the start of the Dixie Fire. As shown in Fig. 6(b), the average clear-sky background temperature cycles between the two stations are very similar during the clear-sky period before the Dixie Fire began,

with a maximum difference of 4° C found in the late afternoon. However, during the BB event on 22 July 2021, station O05 was significantly cooler than station AAT during the daytime, with a temperature difference as large as 10° C in the late afternoon hours. These results indicate that insolation reduction caused by the very optically thick smoke generated a strong surface cooling during the daytime, which could contribute significantly to the observed smoke IR signals at the thermal IR window channels. Still, this ~10° C difference in the 2-m air temperature is smaller than the 23 K difference in MODIS TIR brightness temperature due to dense smoke as shown in Fig. 1(f). This is partly because the brightness temperatures are essentially a surface skin temperature, which is known to be significantly higher than the air temperature under the sunny and dry conditions commonly found in the western United States (Jin and Dickinson, 2010). In addition, the AAT ASOS site is not completely smoke aerosol free and is covered by a thin smoke layer on July 22, 2021 (Fig. 6(a)). Nevertheless, these results show that the 2-m air temperature difference is the largest of the potential causes analyzed in this study.

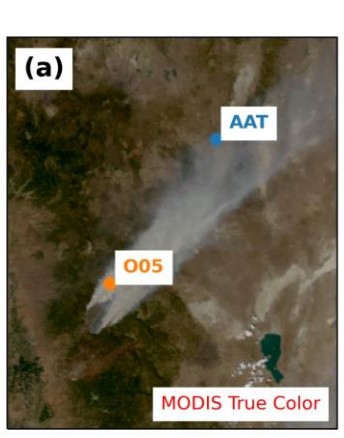
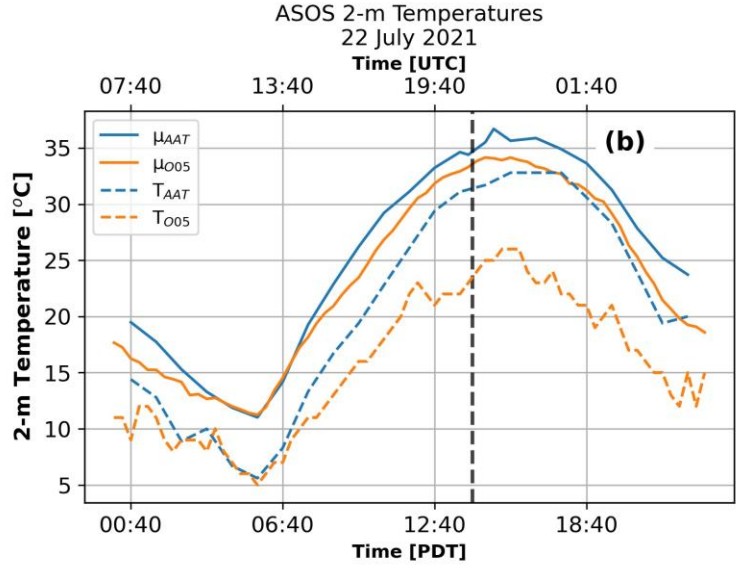

**Figure 6: (a) Aqua MODIS true-color imagery of a smoke plume in northeastern California on 22 July 2021, 21:10 UTC. (b) Climatological average diurnal temperature curves from ASOS stations O05 (solid blue) and AAT (solid orange), calculated using measurements taken between 01 July 2021 and 13 July 2021, are compared to 2-m temperature measurements from OOT (dashed blue) and AAT (dashed orange) taken on 22 July 2021, with the Aqua MODIS overpass time indicated by the vertical dashed black line.**

With the 2-m temperature cooling being observed in the plume region, and with the SBDART results (Fig. 5) showing that the enhanced plume water vapor is not responsible for the TIR cooling, we conduct additional SBDART simulations to test the impacts of the surface temperature on the TOA TIR brightness temperatures. To achieve this goal, we conducted simulations with a focus on altering the surface temperatures, with the CrIS-retrieved surface temperatures at the clear and smoky points being 325 K and 294 K, respectively (Fig. 4(b)). We first conduct a control smoky simulation, in which the surface temperature, as well as temperature and moisture profiles, from the smoky point (dashed blue lines in Fig. 5(f)-(j)) are used as input parameters. The simulated MODIS 11.0 µm / GOES 10.35 µm brightness temperatures for the control smoky simulation are about 25 K below those for the baseline run as shown in the previous section, with the baseline run using CrIS data over the clear point. We also conduct an experimental

smoky simulation using the surface temperature from the smoky point with temperature and moisture profiles from the clear point (dashed purple lines in Fig. 5). Again, a ~25 K difference in simulated MODIS 11.0 µm and GOES 10.35 µm brightness temperature is found between the experimental smoky simulation and the baseline run. Note that the only difference between the baseline run and the experimental smoky simulation is that the surface temperature from the smoky point is used in the experimental smoky simulation. While not shown, an additional simulation was conducted using both the smoky surface temperature and the smoky temperature profile, but the results are nearly identical to those from the experimental smoky simulation, indicating that the temperature of the air column does not impact the simulated TOA TIR brightness temperature; we note that these results are not surprising, as the thermal emission from atmospheric gas constituents is expected to be small compared to the surface emission. This experiment suggests that smoke-induced surface cooling is the primary cause of the observed smoke IR cooling.

Also, as shown in Fig. 4(f), CrIS retrieved air temperatures below the smoke plume over the smoky point are lower than the CrIS retrieved air temperatures over the clear point at the same altitude, possibly caused by shadowing induced by the smoke plume. The cooled temperature profile beneath the smoke plume shows that the thermal effects of the plume are not limited to the surface temperature. Still, the above-mentioned control and experimental smoky simulations suggest that the impact of air temperature cooling due to smoke has a marginal effect on observed TIR smoke cooling as observed from MODIS and GOES.

### 3.4. Nighttime Cross-Check

To further study the impacts of sunlight on the observed TIR cooling signal, and as a cross-check on our previous results, we study TOA brightness temperatures from the 11 µm channel at nighttime. In the absence of sunlight, plume-induced surface insolation reduction is not present, which means observable nighttime TOA IR cooling will be a function of characteristics of the plume itself, including enhanced plume water vapor. A critical aspect of this nighttime analysis is identifying the presence and strength of the BB plume overnight, as a lack of TOA brightness temperature cooling overnight could be a result of plume weakening and/or lack of insolation. While MODIS lacks the ability to visibly detect plumes overnight, the Suomi-NPP Visible Infrared Imaging Radiometer Suite (VIIRS) day/night band (DNB), a panchromatic channel covering visible and near-infrared wavelengths, is capable of detecting visible signals under low-light conditions (Lee et al., 2006). VIIRS also measures thermal IR radiances, which are used to determine if TOA IR brightness temperature cooling is seen overnight. VIIRS TOA IR brightness temperatures are studied in regions of dense smoke overnight, as indicated by regions with high DNB reflectance, to determine if the plume-induced cooling is present overnight.

Figure 7(a) and (d) show the VIIRS 0.67 µm visible reflectance and 10.76 µm IR brightness temperatures on 22 July 2021 at 21:24 UTC, 14 minutes after the Aqua MODIS overpass shown in Fig. 1(a)-(f). Due the close overpass times, both the VIIRS visible and IR imagery appear very similar to the MODIS imagery, with a similar plume region cooling exhibited by the VIIRS TOA IR brightness temperatures. In the VIIRS imagery from the following night at 09:42 UTC 23 July 2021, the VIIRS DNB radiances (Fig. 7(b)) show that, despite changes in atmosphere and plume dynamics in the overnight hours, the smoke remained widespread and thick overnight; however, the associated 10.76 µm brightness temperatures (Fig. 7(e), note the decreased brightness temperature range used on the colorbar in this subplot to show enhanced contrast) show no noticeable signs of locally enhanced cooling in the smoky regions

indicated by the DNB radiances. The following day, 23 July 2021 at 21:00 UTC on, the visible reflectance (Fig. 7(c)) and 10.76 µm brightness temperatures (Fig. 7(f)) reveal that the TOA IR cooling in the dense smoky regions returned under sunlit conditions. This lack of an overnight TOA cooling signal, coupled with the strong daytime TOA cooling signals, suggests that insolation reduction is a key factor behind the observed daytime TOA cooling, and confirms the findings of the radiative transfer model simulations that enhanced plume water vapor effects are not a primary cause

of the cooling at the thermal IR window channel. Figure 7 also indirectly shows that large smoke debris may not be the main cause for the cooling at thermal IR channel either.

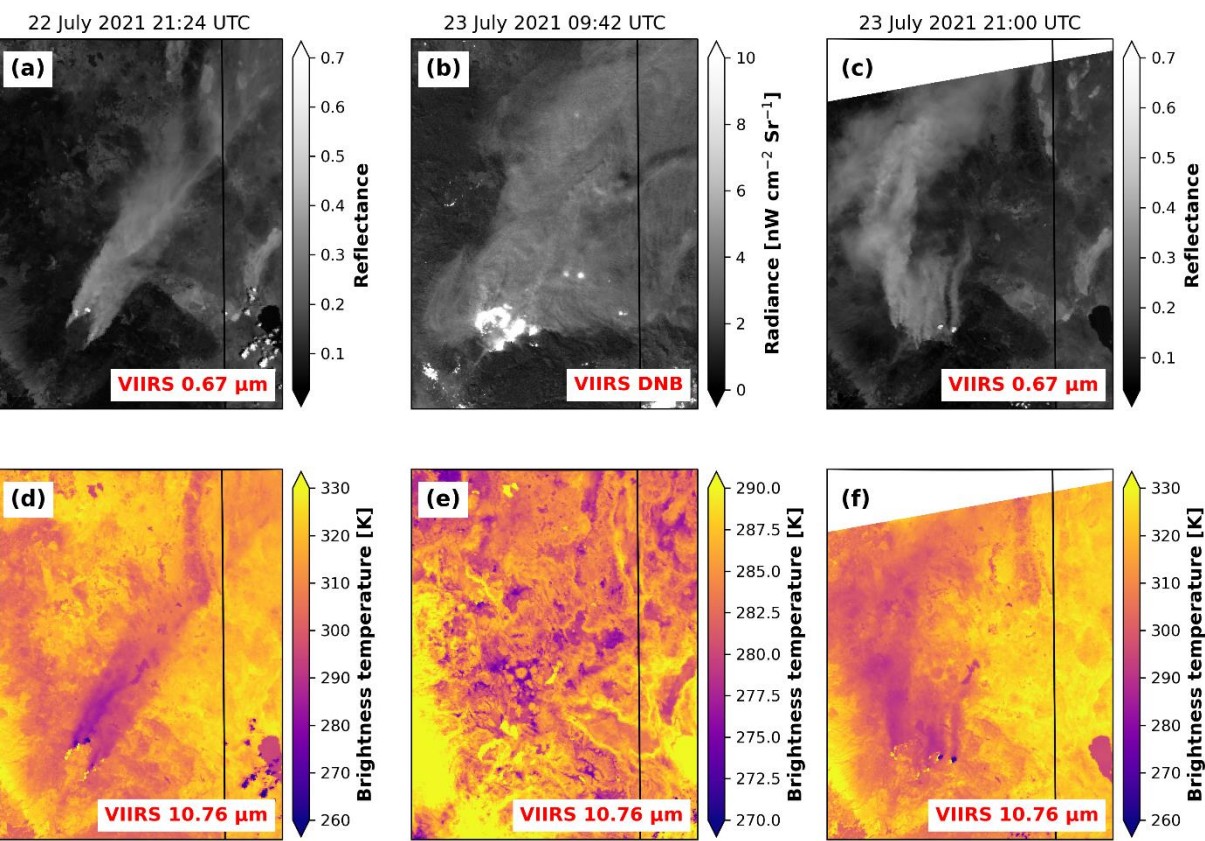

**Figure 7: (a) Suomi-NPP VIIRS visible (0.67 µm) reflectance from the 22 July 2021 21:24 UTC granule. (b) VIIRS day/night band (0.5 – 0.9 µm) radiance from the 23 July 2021 09:42 UTC granule. (c) VIIRS visible (0.67 µm) reflectance from the 23**
**July 2021 21:00 UTC granule. (d) VIIRS thermal infrared (10.76 µm) radiance from the 22 July 2021 21:24 UTC granule. (e) VIIRS thermal infrared (10.76 µm) radiance from the 23 July 2021 09:42 UTC granule. (f) VIIRS thermal infrared (10.76 µm) radiance from the 23 July 2021 21:00 UTC granule. Note the decreased brightness temperature range used in subplot (e) compared to subplots (d) and (f).**

## 4. Radiative balance implications

With this smoke-induced longwave impact observed, our understanding of smoke aerosol radiative balance must be reconsidered. In the past, the direct radiative effects of smoke aerosols over cloud-free skies are estimated at the SW spectrum, as the longwave impacts of smoke aerosols in satellite observations are considered negligible. We re-examine this hypothesis using TOA shortwave (SW) and longwave (LW) flux retrieved from the Cloud and the Earth's Radiant Energy System (CERES) instrument on board the Aqua satellite for the 22 July 2021 21:10 UTC case as

shown in Fig. 1(g)-(i). Over dense smoke regions as identified from the thermal and visible MODIS imagery, the TOA SW flux (SWF) data exhibit significantly increased reflected SW energy (~80 Wm$^{-2}$) compared to nearby relatively smoke-free regions, confirming the shortwave cooling effects of the plume. The observed TOA LW fluxes reveal a similar spatial pattern, with a significant reduction in TOA upwelling LW energy (~ -50 Wm$^{-2}$) in the same region as the increased SWF. Thus, the overall TOA smoke radiative impact (30 Wm$^{-2}$), when considering both the LW and

SW components, is less than half what would be expected assuming that smoke aerosols only have direct impacts in the SW spectrum. This conclusion is supported by the scatter plot of MODIS 11 µm brightness temperature against co-located CERES fluxes, shown in Fig. 8. The CERES footprint is much larger than that of MODIS, so before co-locating the data, all MODIS pixels within the latitude and longitude bounds of each CERES pixel are averaged. A negative relationship between TOA SW flux and MODIS TOA 11.0 µm brightness temperature of – 2 Wm$^{-2}$ K$^{-1}$ is

found for CERES pixels within the plume (Fig. 8(a)), with a positive relationship between LWF and MODIS TOA 11.0 µm brightness temperature of 1.9 Wm$^{-2}$ K$^{-1}$ found in the plume region (Fig. 8(b)). Additionally, while a strong positive relationship exists between CERES TOA total flux and MODIS brightness temperature outside the plume, nearly no relationship exists between the two variables within the plume (Fig. 8(c)). Thus, when considering the TOA longwave radiation effects of these dense smoke plumes in terms of TOA brightness temperature, the total radiative

effect is much more neutral than the commonly held, shortwave only effect. This indicates that the longwave impacts of smoke aerosols need to be considered in future studies for smoke radiative impacts.

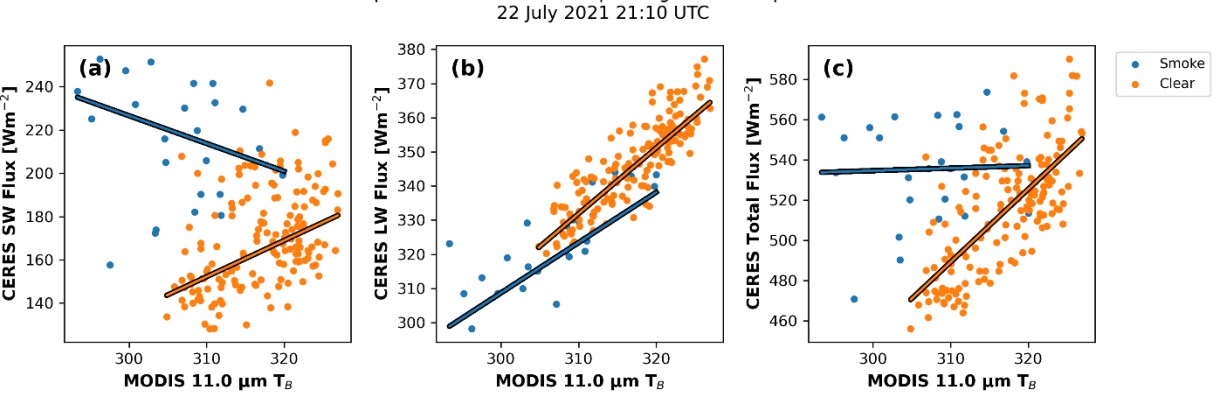

**Figure 8: Scatter plots of CERES SW flux (a), LW flux (b), and total flux (SW + LW, c) with MODIS 11 µm brightness temperature within the plume (blue) and outside the plume (orange) for the MODIS and CERES overpasses on 22 July**
**2021 at 21:10 UTC shown in Fig. 1.**

Lastly, very optically thick smoke plumes pose a difficult obstacle for aerosol retrievals from passive sensors such as MODIS. This is because thick plumes are often misclassified as clouds and thus removed from the retrieval process. We argue that the reduction of brightness temperature at the thermal IR channels may also be used as another indirect measurement of AOD when aerosol optical depth is over the detection limit of the traditional aerosol retrieval methods.

While the work identified a relationship between the increased visible reflectance of the smoke plume (and, therefore, the optical depth) and the magnitude of the cooling beneath the plume, and suggests the potential ability to retrieve AOD of the very dense plume, further work is needed to more effectively remove the cooling impacts of other variables and directly relate the observable TIR cooling to an AOD.

## 5. Conclusions

In this study, we present observational evidence of smoke-induced TOA infrared cooling observed from both polar-orbiting (Aqua MODIS, Suomi-NPP VIIRS and CrIS) and geostationary (GOES-16/17) sensors. While our analysis indicates that coarse particles are not a key factor in causing the TOA IR cooling, we identified co-emitted water vapor in the plume and insolation reduction-induced surface cooling as two causes, with the surface cooling being the primary factor for the IR window channels. The strong longwave cooling response calls into question the long-held

understanding of BB aerosol radiative effects, as the total radiative effect when accounting for the longwave flux reduction is significantly smaller than the radiative effect when accounting only for the increase in shortwave flux. The negative relationship between TOA IR brightness temperature cooling and visible reflectance suggests a relationship between the TOA IR cooling and plume characteristics (i.e. AOD), but further work is needed to investigate the feasibility of retrieving AOD from the TOA IR cooling. Additionally, while this exercise is focused on

studying the TIR characteristics of one fire, we lay out the framework for future, more systematic studies of other wildfire cases.

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

695     study and worked on the data analysis. R. Holz and A. Gumber processed GOES-17 data. W.L. Smith, Sr. processed the CrIS data. All authors worked on writing of the manuscript.

**Competing Interest Declaration:** The authors declare no competing financial interests.

700     **Code Availability:** Python version 3.9 programs were used to analyze the MODIS, CERES, GOES-16/17, CrIS, NEXRAD, and VIIRS datasets studied in this paper. All associated Python programs are available upon request to the author.