# Peer review of "Thermal infrared observations of a western United States biomass burning aerosol plume"

_EGUsphere, 2023_

## Referee Comment (RC1)

Also hydrogen cyanide (HCN) could be a contributor. This was the case for the Indonesian fires of 2015 (see for example https://nwp-saf.eumetsat.int/publications/tech_reports/eresmaa_etal_poster_at_itsc22.pdf

**3.3. Surface radiative response**

[referee-annotated manuscript omitted]

---

## Author Comment (AC1)

**Comment:** The phenomenon on which the manuscript focuses, TIR cooling in apparently "dry" smoke, has been of particular interest to me and others studying energetic fire behavior. I have a few questions and observations to consider.

If I understand the manuscript, the conclusion is that the bulk of TIR cooling is attributable to large AOD of submicron particles. Cooling is characterized remotely with satellite brightness temperature data and in situ with surface weather stations. The satellite-based TIR cooling is as large as 25°C, and the onset is sudden. The in-situ surface-temperature observations show a 1-2C initial cooling followed by a leveling off and thereafter a slow rise until about dusk. Is that a fair characterization? If my understanding is accurate, a question I have pertains to the physical reason ascribed to a supposed sudden surface cooling of up to 25°C. I.e. what would make the surface cool well below its pre-smoke condition? Like any sunny day that is interrupted by a cloud or plume, one might expect an interruption in surface warming, but what mechanism would drive the temperature significantly lower than before the plume started inhibiting insolation. Could you clarify the proposed mechanism for such a dramatic cooling as inferred from the satellite data?

==Response==: Thank you for the comment. The drop in temperature as shown in this study is mostly related to the significant reduction in surface downward solar radiation. This is because downward solar radiation is a key component in maintaining surface energy balance. With the significant reduction of surface downward solar radiation due to the smoke plume, it is not a surprise that we can expect a drop in surface temperature. As with a significantly reduced surface downward SW radiation, and a plausible reduced surface downward LW radiation due to the cooling effect of the atmospheric column above the surface layer due to smoke (as found in this study), to maintain a surface/near surface energy balance, surface upward LW emission, hence surface /or near surface air temperature, must be reduced to maintain the energy balance.

Note that after further analysis of the GOES-17 imagery (and as shown in the figure below), we identified a pyrocumulus cloud that formed over the southeastern side of the fire at approximately 22:30 UTC 20 July 2021, about the same time as that sudden, extreme dip in brightness temperature observed at the orange point (see the bright, white spot in the VIS imagery, as well as the bright spot in the SWIR imagery and the especially cold spot in the TIR imagery above the fire, which is indicated by the bright pixels in the SWIR and TIR). We suspect that this pyrocumulus cloud is likely responsible for the sudden, extreme drop in TIR brightness temperature below the pre-dawn TIR brightness temperature at that point. From the SWIR and TIR observations over the next 15 – 30 minutes (not shown), this pyrocumulus cloud dissipated very quickly after formation, which could explain why the rapid cooling below 280 K was observed at the orange point at about 22:30 UTC and not at the green point farther downwind. Thus, while there was brief pyrocumulus contamination near the fire, strong cooling is still observed at both the orange point and the green points even after the pyrocumulus dissipates, so we can still conclude that pyrocumulus clouds are not primarily responsible for the strong, widespread cooling signal in the plume.

[Figure]

**Figure 1. Comparison of GOES-17 observations (visible reflectance (a), short-wave IR reflectance (b), and thermal IR brightness temperature (c)) at 22:30 UTC 20 July 2021. Red arrows point to pyrocumulus cloud in the smoke plume and above the fire.**

Additionally, as seen in the figure below, we plotted three new time series of the GOES-17 multi-channel observations for points much farther down the plume, and the green point in this figure (which encounters smoke starting at about 23:00 UTC) still sees cooling of about 20 K without the rapid, extreme, and short-lived cooling seen at the orange point in the original figure.

[Figure]

**Figure 2. GOES-17 true color (a), shortwave infrared (2.25 µm, (b)), thermal infrared (10.35 µm, (c)), upper-level water vapor (6.18 µm, (d)), mid-level water vapor (6.95 µm, (e)), and low-level water vapor (7.34 µm, (f)) imagery of the Dixie Fire at 21 July 2021, 00:00 UTC. Third row: Time series of GOES-17 0.64 µm visible reflectance (solid), 2.25 µm shortwave infrared reflectance (dashed), and 10.35 µm brightness temperature (dotted) for points outside of the Dixie Fire smoke plume (blue and orange) and inside the plume (green) downwind of the fire.**

**Comment:** There is excellent NEXRAD coverage of the Dixie fire and downwind area. These data bear directly on this case study. A review of these data reveals that there were radar echoes in the smoke plume far downwind of the fire itself. This indicates large enough particles to impact TIR brightness temperature. The radar data suggest that pyrometeors (a term coined by McCarthy et al., https://doi.org/10.1029/2019GL084305) and/or hydrometeors were in play both on 20-21 and 22 July instances.

==Response==: Thank you for the suggestion. We have compared GOES-17 visible (0.64 micron) reflectance, SWIR (2.25 micron) reflectance, and TIR (10.35 micron) brightness temperature to composite reflectivity derived from the Reno (KRGX) and Beale Air Force Base (KBBX) NEXRAD radars at the same time as the GOES-17 observation, with the comparison for 00:00 UTC 21-07-2021 shown below. Clear returns can be seen in areas just downwind of the Dixie fire, suggesting the existence of pyrometeors and/or hydrometeors in the plume very near to the fire (or may be an indication of Bragg scattering). However, strong GOES-17 TIR cooling signals can still be observed in regions far downwind of the fire, where no returns are shown in the NEXRAD composite reflectivity fields. Thus, we suspect that while pyrometeors and/or hydrometeors may contribute to the TIR cooling signal in regions very close to the fire, they are not primarily responsible for the observed TIR cooling.

[Figure]

**Figure 3. Comparison of GOES-17 observations (visible reflectance (a), short-wave IR reflectance (b), and thermal IR brightness temperature (c)) to composite reflectivity derived from the Beale AFB radar (KBBX, d, southwest of figure) and Reno, NV NWS radar (KRGX, e, southeast of figure) at 00:00 UTC 21 July 2021.**

**Comment:** Although GOES West data were not available for the 22 July case study, GOES East data are. These might offer an opportunity to compare the remotely sensed TIR brightness temperature with radar echoes and the surface station temperature.

==Response==: Thank you for the suggestion. In the figure below, cross sections of the plume as observed from KBBX and KRGX are shown in relation to the GOES-16 VIS, SWIR, and TIR observations at 21:10 UTC 22 July 2021. Note that, due to the very high viewing angle of GOES-16 to the Dixie Fire plume, the visual positions of the station points relative to the smoke plume will vary slightly compared to the GOES-17 and MODIS images, which have smaller

viewing angles. The blue dots in the spatial images show the location of the O05 ASOS site while the orange dot shows the location of the AAT ASOS site. The red dot in the RHI plots indicates the location (distance from the radar and height above sea level) relative to either the KBBX or KRGX radars. From the KBBX PPI and RHI plots, we can see that the O05 ASOS site was located beneath a zone of high reflectivity (> 30 dBZ) and low correlation coefficient (< 0.6). Additionally, the VIS and SWIR imagery show a pyrocumulus cloud located over the western side of the Dixie Fire (see the bright, white cloud in panel A below). Due to its close proximity to the fire, the large region of reflectivity located above the site, and the potential impacts of nearby pyrocumulus at the observation time, additional factors could be influencing the stronger TIR cooling at the O05 site. However, as seen in the KBBX RHI and PPI, the regions with radar reflectivity are primarily close to the fire, while strong TIR cooling is still observed far downwind of the fire, where no reflectivity signals are observed. Thus, this supports our conclusion that pyrometeors and hydrometeors are not primarily responsible for the TIR cooling observed extending far downwind of the fire. We have added some NEXRAD/GOES-17 comparison analysis to Section 3.1 and NEXRAD data description to Section 2 of the paper.

[Figure]

**Figure 4. Comparison of GOES-16 and NEXRAD radar observations derived from the Beale AFB radar (KBBX, southwest of figure) and Reno, NV NWS radar (KRGX, southeast of figure) at 21:00 UTC 22 July 2021. First row: GOES-16 visible reflectance (a), shortwave IR reflectance (b), and thermal IR brightness temperature (c), with radar cross section locations added as red lines along azimuths from KBBX and KRGX and the O05 and AAT ASOS sites indicated by blue and orange dots, respectively. Second row: KBBX plan position indicator (PPI) of composite reflectivity (d), and range-height indicator (RHI) of reflectivity (e) and correlation coefficient (f), with the red dot in (e) and (f) indicating the location of the O05 ASOS site relative to the radar cross section. Third row: as in the second row, but for KRGX. Fourth row: as in the third row, but for a cross section much farther downwind of the fire.**

---

## Author Comment (AC2)

**Reviewer #1 Comments**

We thank the reviewer for her constructive comments. We are unsure as to what the reviewer means by the highlights and check marks in the supplied comment document. We reviewed the wording and science in the lines around the check marks and found no issues with grammar or scientific content. We thus make no changes to the lines with the highlights or check marks (unless specific comments were added in the nearby margins), but we are happy to address any other comments or suggestions that the reviewer may have regarding said un-marked highlights or check marks.

**Comment:** Line 23: (in reference to "… wildfire aerosol plumes are more radiatively neutral…") Due to compensating effects in the VIS and IR?

**Response**: Thank you for the question. That is correct: we suspect that the dense smoke plumes are more radiatively neutral than previously understood because the reduction in upwelling longwave flux related to the TIR cooling offsets some of the increase in upwelling shortwave flux associated with the high albedo of the dense smoke.

**Comment:** Line 169: (at elevations nearly 3000 ft below O05). Metric system

**Response**: Thank you for catching this error. We have changed "… at elevations nearly 3000 ft below O05." to "… at elevations nearly 1 km below O05."

**Comment:** Section 2.7: What is used for the IR biomass burning aerosol optical properties?

**Response**: Thank you for the question. The purpose of our SBDART simulations is only to test if enhanced concentrations of gas constituents (water vapor, carbon dioxide, methane, etc.) could be responsible for the observed TOA thermal IR cooling, so we conducted our simulations by only adjusting the amounts of water vapor and gas constituents in the profile. Smoke aerosol plumes are not included in the SBDART runs.  This is also because observational-based aerosol optical properties are needed.  However, we do not have reliable aerosol observations to constrain the SBDART simulations.

**Comment:** Section 3.2, lines 322 – 335: Also hydrogen cyanide (HCN) could be a contributor. This was the case for the Indonesian fires of 2015 (see for example [link to eumetsat tech report])

**Response**: Thank you for the comment. Hydrogen cyanide certainly could be a contributor. However, no observed data for HCN are available for the study case. Also, we do not observe significant HCN signals at nighttime (Section 3.4).   We revised the paper by adding the following text:

"Note that high concentrations of hydrogen cyanide (HCN) were found for the 2015 Indonesian Fires (Park et al., 2021). However, no observational HCN concentration data are available to confirm the presence of high concentrations of HCN for this study case. Also, if absorption by HCN within the smoke plume plays a significant role in the TIR cooling signal for this study case, we would observe cooling signals within the smoke plume at night, but as we show later in Section 3.4, no significant cooling signal is observed in the plume region at night. We thus expect the impact of HCN to be marginal in this case, but leave further analysis of the impacts of HCN on the Dixie Fire smoke plume to a future study."

**Comment:** Lines 386 – 387 (in reference to "… possibly caused by shadowing induced by the smoke plume.") Does this mean that the smoke acts like a cloud in the VIS but not in the IR? Does this depend on the plume height?

Response: Thank you for the comment. That is correct: our hypothesis is that the sub-micron sized smoke particles act like a cloud in the visible and scatter/absorb significant amounts of sunlight, causing strong shadowing effects on the surface and cooling the surface and column beneath the smoke. However, due to the drastic decrease in smoke optical depth at the IR spectrum, the smoke plumes are likely transparent to IR radiation.

Given the scope of this study, in that we studied just one of these dense smoke plume cases, we are limited in our ability to study if the strength of the IR cooling signal is related to the plume height. Further study is needed to investigate this with the inclusion of lidar observations, which are not available for this study.

**Comment:** Line 470: I would suggest at looking at the Indonesian fires of 2015 – those were extensive and would provide a nice testbed for this hypothesis. GOES data will not be usable over that region but MeteoSat-9 can provide some insight along with the polar orbiting satellites.

Response: Thank you very much for the suggestion. We agree that expanding the analysis to other wildfire cases (including the Indonesian fires of 2015) is an interesting and necessary next step. However, such a step would require extensive additions to this paper, which is already long as it is. Thus, we choose to keep this paper as a first look at the observable smoke-induced TIR cooling phenomenon and leave the application of this hypothesis to other wildfire cases for a future study.

---

## Author Comment (AC3)

**Reviewer #2 Comments**

We thank the reviewer for the constructive comments.

This manuscript discusses the cause of the observed thermal infrared signal of a specific biomass burning episode in western US (the Dixie fires). Different causes are investigated using observations and radiative transfer modelling, and a conclusion is reached that the most important player is the surface cooling by the smoke plume. The data and analysis presented are mostly convincing, although not always straightforward to understand. I would appreciate seeing some additional physics in the discussion (examples given below in the specific comments), and I wonder why CrIS hyperspectral radiance data (in addition to its T and WV profiles, and skin T) is not examined together with the broad TIR channels of GOES and MODIS. That would show pretty nicely (at least it does in the IASI data I have looked at) how the TIR radiance is "flatly reduced" by about 25K for smoky observations, while the (low intensity) gas absorption lines are not toomuch affected. Such a flat reduction of atmospheric window TIR "baseline" BT directly points to a (skin) surface temperature reduction or a thin cloud (aerosols usually have specific signatures with slopes and/or different impacts on radiance at about 9 and 11μm - see for example Clarisse et al DOI 10.1364/AO.49.003713). The trick, I think, in this case, is to ensure that the observed lower BT is due to lower surface temperature and not to thin (water) clouds (ice clouds would have a typical TIR signature which is not observed). This is where combining the TIR with SWIR and visible observations comes in.

**Comment:** suggestion: mention the episode name in the paper title - it was important enough to even have a wikipedia page ;)

Response: Thank you for the suggestion. While we did consider including "Dixie Fire" in the paper title prior to submission, we chose to leave the title as "western US biomass burning aerosol plume" since we found that this behavior can be seen in many other cases of large wildfires from the western US.

**Comment:** lines 23-25: this sentence is rather unclear to me. I guess the authors means the very thick optical depth at visible wavelength? I am also very puzzled as to how a BT change in TIR could be mapped to a smoke signal, especially since the authors highlight that the smoke signal is due to surface cooling - I mean how would the difference be done between smoke and any other surface cooling reason, such as a cloud?

Response: Thank you for the question. That is correct: we mean the very thick visible optical depth. Traditional passive-based aerosol-retrieving algorithms struggle in very high-AOD situations (due to the misclassification of thick smoke plumes as clouds), but we hypothesize that if the strength of the surface cooling caused by the smoke-induced surface shadowing is a function of the visible optical depth of the smoke plume, it may be possible to retrieve AOD

information from the surface cooling in those regions where the current algorithms struggle. Significant amounts of work would indeed be required to quantitatively make that connection and remove uncertainties (including screening out impacts from cloud and background meteorological changes), but the focus of this paper is to show that a significant TIR cooling signal from dense smoke plumes can be observed from multiple satellite platforms and the signal can be traced to surface cooling beneath the smoke.

**Comment:** lines 36-39: in addition to the scattering, the aerosol absorption plays a role in the TIR (usually a dominant role) and this can happen no matter the particle size - of course the particle number concentration needs to be higher for smaller particles to have an observable signal, and the particles need to be absorptive in the TIR

**Response**: Smoke aerosols are typically transparent at the IR spectrum from satellite observations, indicating the aerosol optical depth of smoke aerosols (including absorbing optical depth) is small at the TIR spectrum. Also, we would observe smoke TIR signals at night if aerosol absorption plays a significant role at the TIR spectrum. However, smoke aerosol signals are much less observable at night than during daylight hours. Thus, we expect the impact of smoke aerosol absorption at TIR spectrum to be marginal. Still, it is an interesting topic to explore and we leave this topic for a future study.

**Comment:** section 2: I suggest to re-define all acronyms within each sub-section here - but this is just a suggestion and may also be left to the editorial staff

**Response**: Thank you for the suggestion. We choose to leave this to the editorial staff to make the decision.

**Comment:** line 128: CONUS domain?

**Response**: Thank you for the note. We have added a definition for the CONUS acronym (contiguous United States).

**Comment:** line 133: 12Z and 03Z?

**Response**: Thank you for the comment. "12Z" and "03Z" refer to "12 Zulu" and "03 Zulu", respectively, with "Zulu" being a common way to refer to UTC time.  We replaced "Z" with "UTC" for clarity.

**Comment:** section 2.4 considering that the observations by CrIS are affected by the smoke, and that this is not accounted for in the CrIS retrieval algorithm (at least I would guess it is not), how

confident may we be in the retrieved surface and atmospheric temperature and atmospheric humidity for the smoky observations?

Response: Thank you for the question. The reviewer's question is an important one which requires further study to support the answer provided here. The answer provided here is based on the data provided in this study and the published results of an independent study (Zhou et al., 2021). The conclusion from the independent study is based on airborne (NASA ER-2) hyperspectral radiance measurements, like those obtained with the satellite CrIS instrument, and co-located LIDAR data obtained during the FIREX-AQ field campaign conducted over the western US during August 2019.

**A. Conclusions Based on the Dixie Fire Study:**

The analysis of MODIS 2.1 reflectance and GOES-17 TIR (indicate significant thermal cooling below regions of relatively high NIR (2.25 micron) reflectance of the smoke plume. However, much of his TR cooling is believed to be due to the smoke induced reflectance shading of the atmosphere and surface beneath the plume. As shown in figure 1 below, the thermodynamic retrievals obtained from the CrIS radiance spectra (Smith et. al., 2012), provide consistent differences with the independent Rapid Refresh (RAP) 2-hour forecast atmospheric profiles, within (blue lines) and outside (orange and green lines) the smoke plume. These results indicate that the satellite retrievals (see figure 1 below) within the smoke plume are not being affected significantly by smoke particles. It is noteworthy that the retrieved cooling within the boundary layer near the surface, due to the plume sunlight reflection, is not captured by the RAP forecast, which apparently has not accounted for the Dixie Fire smoke plume in the model forecast.

[Figure]

**Figure 1.** Retrieval Vs RAP 2-hr Forecast for the orange, blue, and green locations shown in Figure 3 of the manuscript. The retrieval values are denoted by the solid lines whereas the RAP forecast values are denoted by the dashed lines.

Also, the retrieved near-surface air temperature (295.4 K) and surface skin temperature (308.2K) shown in figure 2 below is consistent with the 2-m surface temperature of 23C (296.2 C) observed under the Dixie fire plume at station O05, as shown in figure 6 of the manuscript. Note that the surface skin temperature retrieved at this point is about 10 K warmer than the nearby-observed (O05) and retrieved near- surface air temperatures.

[Figure]

**Figure 2.** Surface-Skin and Near-surface Air Temperature retrieved from CrIS cloud-free radiance observations. The circled values are for the retrievals closest to the Roger's Field, Chester California, 2-m surface observation location. Note that the near-surface air temperature is the temperature profile retrieval value at the lowest profile level above the surface.

Considering that the retrieved profile discrepancies with RAP forecast profiles do not show any systematic dependence on the retrieval location relative to the smoke plume and that the retrieved near-surface air temperature under the smoke plume is in relatively good agreement with the observed 2-m surface air temperature, the retrieved surface-skin temperature being considerably warmer, it is concluded that the smoke particles have little radiance attenuation impact on the profile retrievals in this case.

**B.  Conclusions Based on FIREX-AQ Airborne Retrieval and LIDAR Fire Observations:**
Results of an independent study (Zhou et al., 2021) utilizing airborne hyperspectral radiance measurements (NAST-I), which are like the radiance measurements with the satellite CrIS instrument, and co-located LIDAR (CPL) data, obtained during the August 2019 FIREX-AQ airborne campaign, support the conclusion stated above. In this reference paper, it is shown that the retrieval of the atmospheric humidity profile has little unexpected dependence on the density of smoke particles observed by the LIDAR and that the retrieved CO concentration profile coincides with the smoke plume particle density as expected (figure 8 of the reference).  Most important, the retrieved CO retrieved values are in relatively good agreement with in-situ observations (figure 7 of the reference).  Thus, this study supports the conclusion drawn in section A above, that hyperspectral infrared retrievals of humidity and trace gas profiles are not influenced significantly by smoke particle attenuation of the observed upwelling spectral radiance.

**References:**

Zhou DK, Larar AM, Liu X, Noe AM, Diskin GS, Soja AJ, Arnold GT, McGill MJ. Wildfire-Induced CO Plume Observations From NAST-I During the FIREX-AQ Field Campaign. IEEE J Sel Top Appl Earth Obs Remote Sens. 2021; 14:2901-2910. doi: 10.1109/jstars.2021.3059855.

Smith., W. L., E. Weisz, S. V. Kireev, D. K. Zhou, Z. Li, and E. E. Borbas (2012), Dual-Regression Retrieval Algorithm for Real-Time Processing of Satellite Ultraspectral Radiances, J. Appl. Meteor. Climat., 51, 1455-1476, doi:10.1175/JAMC-D-11-0173.1.

**Comment:** section 3.1: I would appreciate here some introduction about the expected effect of the studied phenomena (coarse / giant particles, pyrocumulus) on the radiance in the different channels used in this work; that would allow the reader to follow the developments and understand the conclusions more easily

**Response:** Thank you for the suggestion. We have added discussion to explain the expected impacts of the large particles on the observed radiances and reflectances.

**Comment:** figure 2: what are the Z after the hours? (as line 133)? Is this local time, or UTC, or other?

**Response:** Thank you for the question. As mentioned above, "Z" refers to "Zulu" time, which is a common way of referring to "UTC" time. Nevertheless, we have replaced the "Z" in the figure with "UTC" for clarity.

**Comment:** section 3.2: Again here I would appreciate an introduction about the signatures of gases in the TIR atmospheric window, especially the fact that they are rather small; maybe using cross-sections also. Within the GOES 10.35µm and MODIS 11µm channels, one would indeed find some weak WV absorption bands, some very weak $CO_2$ absorption, relatively intense $O_3$ absorption, possibly $NH_3$ absorption if some is present but no $N_2O$ or $CH_4$ absorption (no band in those channels). Because in that spectral range gas absorption is rather low, no increase within physical range of any of those gases would lead to 25K BT reduction at 11µm. This is also the conclusion that the authors reach after experimentally testing that hypothesis, and the proposed approach is also important and interesting but I think that the physical base should be discussed as a complement.

**Response:** Thank you for the suggestion. We have added discussion to section 3.2 to describe the absorption lines of each of the gas constituents in the TIR spectrum.

**Comment:** lines 295-298: could you elaborate on what might be the reason for this difference? Would this be a sign that the CrIS "smoky" profiles are not "close to reality" because the smoke impacts the observation and this is not accounted for in the retrieval?

**Response:** Thank you for the question. There are many potential reasons for these differences, including uncertainties in the CrIS retrievals, but also from the simplistic nature of the SBDART simulations. In these simulations, we check only if the temperature and water vapor profiles in the smoke plume region are responsible for the observed TIR cooling signal, so we lack information about other gas species that could affect the water vapor channel signals. Additionally, we do not include any aerosol information in the SBDART simulations, which could also impact the comparison between the simulated and observed values. Thus, while there are several possible explanations for these differences, we cannot definitively state why the differences exist.

**Comment:** figure 4 (j): where is the orange line? I would guess exactly behind the green but I would mention it in the legend, or redo the plot with e.g. different line widths so that the line can be seen.

**Response:** Thank you for the note. We have widened the orange lines to make them more visible when they lie directly beneath the green lines.

**Comment:** section 3.3: again a bit of physical basis here would be useful, I think. The expected BT in channels around 11μm, in absence of clouds / aerosols (and at relatively low viewing angles), would be slightly lower than the surface skin temperature (unless looking above a surface with low emissivity, of course). This is indeed what the radiative transfer shows when using either the CrIS skin T from "clear" or "smoky" cases (but was this CrIS skin T "right" for the smoky pixel?). One question that remains non-addressed in this section would be if it is reasonable that the surface (skin T) would cool by 25K in a short time if under a thick smoke plume. I guess this could be done by looking at other days (without smoke) BT daily cycles and how much night BT differs from day BT under relatively similar circumstances (except the smoke), and how long the surface takes for cooling after sunset, for example.

**Response:** Thank you for the comment. We do not feel that comparing the diurnal temperature variation of the same area under a clear-sky case to the cooling in the smoky case would be valid, since on a clear-sky day the amount of incoming sunlight gradually decreases through sunset, but in this case there is a sudden "shutting off" of the incoming sunlight at the surface. In a way, this "control case" of clear-sky diurnal cooling speed can be seen from the blue dot in the figure, which is never under smoky or cloud conditions during the latter portions of the study day, so the timing of the gradual cooling can be seen there.

**Comment:** figure 5 (b): time is here given in local time while I think almost everywhere it was UTC time. This should be consistent and my preference would be to have both UTC and local time on each plot (local is interesting to know which part of the day we are looking at, while UTC time is interesting if the reader wants to compare with any other data)

Response: Thank you for the suggestion. We have added UTC time as a second x axis on the top of the time series.

**Comment:** lines 382-383: again some physical explanation here would be nice - indeed the atmospheric temperature has only a second order impact on the observed BT, being through the atmospheric gases thermal emissions, which depend on their temperature and cross-section - the latter being rather low in the used channels

Response: Thank you for the comment. We have added discussion accordingly.

**Comment:** line 410 and following: I am not so sure one can say that there is no noticeable cooling in the plume area - the plume is widespread and there's some widespread "reddish" area in VIIRS 10.76μm that seems to match the grey area in VIIRS DNB. However this "feeling" might come mostly from the fact that a completely different BT scale is used for that plot with respect to the 2 "day" plots. I would strongly suggest using the same scale, or at least the same range of temperatures for the color scale (currently 70K for daytime and 20K for nighttime)

Response: Thank you for the comment. We initially used the same BT scale across all VIIRS 10.76 um plots, but the contrast was far too low in the nighttime plot to definitively say if there is any nighttime TIR cooling in the plume region (see the original version of the figure below, which uses the same color bar across all three 10.76 um BT plots). Thus, we chose to use a smaller BT scale for the nighttime imagery to enhance the contrast and make it easier to determine if the smoky region in the nighttime plots are warmer or cooler than the surroundings. The enhanced contrast also allows one to clearly identify surface features in the nighttime TIR imagery from beneath the smoke plume. If the smoke particles themselves were causing any nighttime TIR cooling signal, we would expect any surface features to become obscured by a uniform region of decreased temperatures; thus, since clear surface features are visible in the nighttime imagery, the smoke particles themselves are not causing TIR cooling at night.

[Figure]

**Figure 3. Suomi-NPP VIIRS visible (0.67 μm) reflectance from the 22 July 2021 21:24 UTC granule. (b) VIIRS day/night band (0.5 – 0.9 μm) radiance from the 23 July 2021 09:42 UTC granule. (c) VIIRS visible (0.67 μm) reflectance from the 23 July 2021 21:00 UTC granule. (d) VIIRS thermal infrared (10.76 μm) radiance from the 22 July 2021 21:24 UTC granule. (e) VIIRS thermal infrared (10.76 μm) radiance from the 23 July 2021 09:42 UTC granule. (f) VIIRS thermal infrared (10.76 μm) radiance from the 23 July 2021 21:00 UTC granule. This figure is the same as Figure 6 in the paper, but with the same colorbar range used for all three VIIRS thermal IR plots.**

**Comment:** lines 434-436: is this also true at night?

**Response**: Thank you for the suggestion. We do not expect this relationship to hold at night, as Figure 6e shows that there is no distinguishable longwave cooling signal within the plume region compared to the nearby clear regions, and we anticipate that CERES LWF measurements will exhibit similar behavior to the TIR brightness temperatures (for example, see the similarities between the MODIS TIR brightness temperatures shown in Fig. 1f and the Aqua CERES LWF measurements shown in Fig. 1h). Nevertheless, we analyzed nighttime NOAA-20 CERES fluxes over the same region for the satellite's nighttime overpass on 2021-07-23 and found no distinguishable warming/cooling pattern in the plume region relative to the surrounding regions, which verifies that the longwave behavior exhibited during the daytime comparison does not hold at night.

[Figure]

**Figure 4. NOAA-20 CERES TOA LWF observations around the Dixie Fire at night, during the hour of 10:00 UTC 23 July 2021.**

**Comment:** lines 439-441: those numbers are rather different from the numbers given lines 430-436: +80 -50 W/m2 do not compensate, while -2 and +1.9 W/m2K almost exactly compensate. Am I missing something? Or is this within error margin?

Response: Thank you for the question. The first numbers represent the general difference between the SWF/LWF values in the clear region versus the smoky region, while the second numbers represent the slope of the change in forcing with respect to brightness temperature (the slopes of the trend lines fitted in Figure 7). You are correct that the two slopes do indeed compensate, causing the slope of the trend line fitted to the summed SWF/LWF values in Figure 7c to be near zero, but the general total flux values in the smoky region are overall slightly larger than the total flux values in the clear region.

**Comment:** Figure 7a (and c to a smaller extent): I find it rather hard to really see the linear relationship in these clouds of points. In Figure B it is much more clear. Are you sure that a linear relationship is expected between the SW flux and the TOA LW 11μm BT?

Response: Thank you for the question. While there certainly are outliers in the clouds of points, partially due to limitations in the algorithm used to select the smoky MODIS pixels, we find that the linear relationship works well enough to reflect the behavior of the SWF within the smoky and clear regions, which can be seen in Figure 1. The biggest takeaway from the figure is that there is no real relationship between MODIS TIR brightness temperature and CERES total flux

in the smoke plume region, while there are at least weak relationships between MODIS TIR brightness temperature and CERES SWF and LWF.

**Comment:** lines 455-457: this sentence is a bit too straightforward and maybe misleading. The characteristics of the plume can not be retrieved based on the TIR channels, at least this is not what the manuscript is about. Maybe the authors could say that plumes could be identified from the observed BT changes in the TIR, after additional work allows discriminating the reason for those BT changes (clouds, Ts changes, smoke).

Response: Thank you for the suggestion. We have modified the sentence to indicate that a possible avenue of future research is to expand this analysis to possible AOD retrieval, after much additional work is done to clean the signal.

---

## Author Comment (AC4)

**Reviewer #3 Comments**

**Comment:** This is a very interesting study focused on the longwave (LW) signatures of a thick smoke plume from the 2021 Dixie Fire in California. The satellite remote sensing aspects, radiative transfer modeling, and surface station analysis are generally well constructed. The narrative is well-written and organized. My only major comment relates the coarse and giant particles analysis in Section 3.1, which is critical to the conclusions of the paper.

I agree with the comments by other reviewers on incorporating weather radar data to provide a better representation of the horizontal and vertical extents of large smoke debris (ash and other pyrometers). The Dixie Fire plume should have good coverage from radar sites in the region. Adding these data into the current analysis will provide a better constraint on variables driving the observed cooling.

Reflectivity and correlation coefficient (CC) provide a quick and definitive way to examine the presence (or lack thereof) of both hydrometeors (high CC) and pyrometeors (low CC) in the plumes. Smoke plumes examined with radar in previous studies coincided with radar echoes 20+ km downwind, indicative of large particles far from the fire (e.g., McCarthy et al, Lareau et al, Peterson et al.; see example papers below). A quick check of the meteorology on the dates examined here reveal relatively strong winds in the mid-troposphere, which would likely facilitate transport of these larger particles, perhaps as far as the ground station sites. In addition, the analysis period appears to coincide with very intense fire behavior, which would result in higher altitude injections of larger pyrometeors.

https://doi.org/10.1029/2019GL084305

https://journals.ametsoc.org/view/journals/bams/103/5/BAMS-D-21-0199.1.xml

https://journals.ametsoc.org/view/journals/bams/103/9/BAMS-D-21-0049.1.xml

Response: We thank the reviewer for the constructive comments. We further study the potential impacts of large pyrometeors and/or hydrometeors on the observed TIR signal by comparing WSR-88D radar data to the GOES-17 observations. Two nearby NOAA WSR-88D radars, KBBX (Beale Air Force Base, southwest of the Dixie Fire) and KRGX (Reno, Nevada, southeast of the Dixie Fire) provided good coverage of the smoke plume area, so horizontal and vertical cross sections of reflectivity and correlation coefficient at 2021 July 21 00:00:00 UTC are analyzed. Three cross sections of the radar data through the plume region are analyzed, with the cross sections plotted over the GOES-17 visible reflectance, SWIR reflectance, and TIR brightness temperature in Fig 1a, b, and c (below). The PPI of reflectivity from KBBX (Fig. 1d, below) shows regions of high (> 20 dBZ) reflectivity in the regions of the smoke plume immediately downwind of the fire, but extending no more than 20 km downwind of the beginning of the plume. The KRGX PPI reflectivity (Fig 1g, below) shows high reflectivity in

similar regions near the fire, but also has regions of low reflectivity extending farther downwind than the KBBX observations show. RHI cross sections of reflectivity through the plume region along the 36 degree azimuth from KBBX (Fig 1e-f, below) show a column of high reflectivity (maximum of 40 dBZ) and very low correlation coefficient (< 0.6) centered about 70 km away from the radar and extending up to 6 km above the radar, with moderate reflectivity and slightly higher (~0.7) correlation coefficient observed at lower heights to about 90 km away from the radar. The high reflectivity and low correlation coefficient suggest the presence of pyrometeors in this region of the plume (although we cannot rule out the impacts of Bragg scattering, for which we have no data to confirm).

However, a cross section of the plume from KRGX very far downwind of the plume, in regions that the GOES-17 visible reflectances show large amounts of smoke and the GOES-17 TIR brightness temperatures show strong cooling, show next to no reflectivity. The same magnitudes of TIR cooling are observed far downwind of the fire, where there are no radar returns, and very close to the fire, where there are significant radar returns. While the KRGX radar is at a much higher elevation than the KBBX radar (2950 m AGL for KRGX, 67 m AGL for KBBX) and thus may not see large ash and/or BB smoke particles below the radar level, even the KBBX radar does not observe any returns far downwind of the fire, as indicated by both the KBBX PPI and RHI diagrams.

While the KBBX and KRGX reflectivity and correlation coefficient observations may suggest the presence of pyrometeors in the plume in close proximity to the fire (and, again, we cannot rule out possible Bragg scattering), the GOES-17 SWIR reflectances do not show significant increases in reflectance in those same regions, which would be expected if large (> 1 um) particles were present in the plume. We thus cannot conclusively state if pyrometeors were present in large amounts near the plume region. Regardless, with the same magnitude of strong TIR cooling being observed in regions with no radar reflectivity and with high reflectivity, we conclude that pyrometeors and hydrometeors are not the primary cause of the TIR cooling signal. Additional factors must be in play.

We have added this analysis to the paper under Section 3.1.

[Figure]

**Figure 1. Comparison of GOES-17 and NEXRAD radar observations derived from the Beale AFB radar (KBBX, southwest of figure) and Reno, NV NWS radar (KRGX, southeast of figure) at 00:00 UTC 21 July 2021. First row: GOES-17 visible reflectance (a), shortwave IR reflectance (b), and thermal IR brightness temperature (c), with radar cross section locations added as red lines along azimuths from KBBX and KRGX. Second row: KBBX plan position indicator (PPI) of composite reflectivity (d), and range-height indicator (RHI) of reflectivity (e) and correlation coefficient (f). Third row: as in the second row, but for KRGX. Fourth row: as in the third row, but for a cross section much farther downwind of the fire.**